# Brief communication: Increased glacier mass loss in the Russian High Arctic (2010-2017)

Christian Sommer[1], Thorsten Seehaus[1], Andrey Glazovsky[2], Matthias H. Braun[1]

[1]Institut für Geographie, Friedrich-Alexander-Universität Erlangen-Nürnberg, Erlangen, 91058, Germany
[2]Institute of Geography RAS, Moscow, 119017, Russia

*Correspondence to*: Christian Sommer (chris.sommer@fau.de)

**Abstract.** Glaciers in the Russian High Arctic have been subject to extensive atmospheric warming due to global climate change, yet their contribution to sea level rise has been relatively small over the past decades. Here we show surface elevation change measurements and geodetic mass balances of 93% of all glacierized areas of Novaya Zemlya, Severnaya Zemlya and Franz Josef Land using interferometric synthetic aperture radar measurements taken between 2010 and 2017. We calculate an overall mass loss rate of -22±6 Gt a$^{-1}$, corresponding to a sea level rise contribution of 0.06±0.02 mm a$^{-1}$. Compared to measurements prior to 2010, mass loss of glaciers on the Russian archipelagos has doubled in recent years.

## 1 Introduction

The Arctic has undergone large environmental changes due to polar climate change (Box et al., 2019). An increase in glacier mass loss has been observed in many polar regions (Morris et al., 2020; Zheng et al., 2018; Ciracì et al., 2020). The Russian High Arctic, including the archipelagos Novaya Zemlya, Severnaya Zemlya and Franz Josef Land, is one of these regions. Despite a glacierized area of ~52,000 km², in-situ observations of glacier mass change are sparse. Previous regionwide assessments were mostly limited to the early 21$^{st}$ century and based on gravimetry (Gardner et al., 2013; Jacob et al., 2012; Matsuo and Heki, 2013; Wouters et al., 2019) and altimetry (Ciracì et al., 2018; Moholdt et al., 2012). Most of these studies show mass change rates ranging from -5 to -10 Gt a$^{-1}$. However, both methods have limitations: altimetry requires spatial interpolation while uncertainties of the gravimetric approach might arise from the scattered ice caps and various corrections related to surrounding oceans, surface hydrology and glacial isostatic adjustment (GIA). In this study we have measured surface elevation changes of most Russian Arctic glaciers from digital elevation models (DEM) to derive geodetic mass changes between 2010 and 2017. We use synthetic aperture radar (SAR) DEMs of the TanDEM-X satellites which are independent from cloud cover and provide a high spatial resolution. However, the SAR data derived elevation change rate can be biased by differences in signal penetration depth into the glacier surface between DEM acquisitions of different seasons. The depth of signal penetration is related to the prevailing glacier surface conditions at the acquisition time. In general, SAR penetration is close to zero for melting snow surfaces and bare glacier ice and increases in dry snow. X-band penetration depths of several meters have been observed in different regions (e.g. Millan et al., 2015; Zhao and Floricioiu, 2017; Abdullahi et al., 2018; Li et al.,

2021). Previously, penetration depths have been estimated by a number of studies (e.g. Abdullahi et al., 2018, 2019; Li et al., 2021), using backscatter intensity. SAR backscatter intensity depends on physical properties of the glacier ice, such as grain size and density, roughness and water content (Wessel et al., 2016) and changes between melting and frozen conditions. Thus, we apply a regional correction for relative differences in SAR penetration, based on backscatter intensity, to account for different TanDEM-X acquisition periods of the Novaya Zemlya ice cap.

## 2 Data & Methods

### 2.1 SAR data and penetration depth estimation

Interferometric SAR DEMs are acquired by the bistatic TerraSAR-X add-on for Digital Elevation Measurement mission (TanDEM-X), operated by the German Aerospace Center and Astrium Defence and Space. The TanDEM-X DEMs provide an almost complete coverage of the Russian Arctic archipelagos but can suffer from differences in SAR signal penetration depth

into the glacier volume. When subtracting elevations of SAR DEMs from different seasons, the depth of signal penetration might differ between acquisitions, due to changing surface conditions, and bias the elevation change rate. The TanDEM-X data over most of our study area were acquired during winter 2010/11 (94% of total glacier area) and winter 2016/17 (83%) at temperatures well below zero degrees Celsius and frozen ice surfaces. It is likely that for those acquisitions the difference in X-band penetration depth is small as the SAR data was acquired in the same season and the presence of surface melt or liquid

water is very unlikely in the Arctic winter months. However, for some glacier areas of Novaya Zemlya (35%) and Franz Josef Land (6%), SAR data from September 2016 had to be included to calculate elevation changes because there were no winter scenes available. Using those DEMs without further correction can bias the measured surface elevation change as seasonal changes in snow and ice properties of the glacier surface have significant impacts on the SAR penetration depth (Abdullahi et al., 2019). Fig. 1a shows the hypsometric backscatter distribution of TanDEM-X acquisitions between September 2016 and

January 2017 on Novaya Zemlya. While average backscatter intensity of the October-January acquisitions shows similar patterns, a clear difference is observed for the September DEMs. At altitudes above ~400 m a.s.l., the September data show much lower backscatter values than the respective winter scenes, indicating different surface conditions at the acquisition times. To estimate the difference in penetration depth between the September 2016 and winter 2016/17 DEMs, we derive the measured elevation difference and respective backscatter intensity from overlapping glacier areas which were covered by the September

as well as winter acquisitions. Those reference areas cover a total glacier area of ~2,500 km² and are equally distributed across the Novaya Zemlya ice cap (Fig. S2a). The measured absolute vertical offsets of those glacier areas $\Delta h_{W\text{-}A}$ (Fig. S2c) are then converted to penetration lengths into the glacier volume $l_p$ (supplement section 3.1 & 3.2).

Thereafter, $l_p$ is aggregated and compared to altitude and backscatter intensity of the September acquisitions, respectively. As shown in Fig. 1c and Fig. 1d, the offset between the September and winter 2016/17 backscatter intensity increases above

elevations of ~400 m a.s.l. while the difference in estimated signal penetration increases with higher backscatter intensities of

the September acquisitions. Based on this relationship between penetration difference, backscatter intensity and altitude, a linear regression model can be created to estimate the length of penetration $l_p$ (EQ. 1):

$$l_P = \beta_0 + \beta_1 \times Int \quad \text{EQ. 1}$$


*Int* is the backscatter intensity in decibel and $\beta_0$ and $\beta_1$ the regression coefficients. To fit $l_p$ and *Int*, the difference between the September and winter TanDEM-X acquisitions and backscatter intensity of all overlapping DEM pixels is used. To predict the bias in surface penetration between the September and winter acquisitions 2016/17, the model is then applied to all glacier areas above 400 m a.s.l. on Novaya Zemlya which were only covered by September 2016 SAR (~7,800 km², Fig. S2a). Even-

tually, the estimated surface penetration lengths are converted back to vertical differences in elevation by rearranging the respective equations (supplement sections 3.1 & 3.2). The predicted vertical correction values (Fig. S2b) are then added to the September 2016 elevations and the corrected elevation change rate is calculated. We did not adjust for differences in incidence angle or effective baseline because the viewing geometries of the majority of the used SAR acquisitions are rather similar (Table S2). For 99% of the glacierized area of Novaya Zemlya, the difference in incidence angles is not larger than 2° (39.3°

- 41.3°) while for 93% of area the average baseline is 91.9 m (87.8 m – 95.4 m).

For Franz Josef Land and Severnaya Zemlya, the elevation change rate is not corrected, as on both archipelagos the temporal offset between DEM acquisitions is much smaller than on Novaya Zemlya. Average backscatter intensity is relatively homogeneous for all 2016/17 acquisitions on Severnaya Zemlya (Fig. S1c) while on Franz Josef Land only a very small fraction of September TanDEM-X acquisitions (6%) shows significant differences in backscatter intensity (Fig. S1a). Therefore, trans-

ferring the empirical relationship between differences in surface penetration and September backscatter intensities on Novaya Zemlya to those archipelagos would rather increase the uncertainty of the elevation change measurement.

### 2.2 Glacier elevation- and geodetic mass change calculation

Glacier elevation change rates are calculated by differencing TanDEM-X DEMs of different acquisitions. For the Russian Arctic, TanDEM-X acquisitions of winter 2010/11 (Dec-Feb, Apr) and autumn/winter 2016/17 (Sep-Feb) are available. Ele-

vation models are derived from TanDEM-X Co-registered Single look Slant range Complex (CoSSC) data, closely following the workflow of Braun et al. (2019) and Seehaus et al. (2019). A detailed description of the interferometric DEM generation, co-registration and uncertainty assessment is provided in the supplement. Eventually, the co-registered TanDEM-X DEMs are merged to create two elevation mosaics of winter 2010/11 and 2016/17 and differenced to derive glacier elevation and volume change rates based on glacier areas of the Randolph Glacier Inventory (Pfeffer et al., 2014). Glacier volume changes are

converted to mass change using two density scenarios. For a) a conversion factor of 850±60 kg m$^{-3}$ (Huss, 2013) is applied and for b) 900±60 kg m$^{-3}$ as an approximation of the density of ice. For Novaya Zemlya, the geodetic mass change rate ($\Delta M/\Delta t$) is calculated using the uncorrected elevation change rate ($\Delta h/\Delta t$ $_{uncorr.}$), as derived from the DEM differencing, as well as the surface penetration corrected elevation change ($\Delta h/\Delta t$ $_{corr.}$). Additionally, glacier elevation changes are derived specifically for

marine- and land-terminating glaciers (Fig. S3), following the glacier terminus classification of the Randolph Glacier Inventory.

## 3. Results

For the DEM acquisitions of 2016/17 on Novaya Zemlya, a distinct difference in backscatter intensity is visible between SAR data acquired in September 2016 and October-January 2016/17 (Fig. 1a). The average vertical difference in surface elevation on the respective overlapping glacier areas (Fig. S2a) is 2.13 m. Also, the elevation change rates derived from all glacier areas which were acquired in September 2016 (Fig. 1b), show an average difference in surface lowering of 0.4 m a$^{-1}$ compared to areas acquired in winter 2016/17. Furthermore, the elevation change rate of the period winter 2010/11 to winter 2016/17 is consistently more negative at all altitudes while the change rate between winter 2010/11 and September 2016 indicates elevation gains at the highest glacierized altitudes. The analysed vertical elevation differences of the overlapping glacier areas (Fig. S2a) and the respective backscatter intensity of the September datasets (Fig. 1a) indicate altitudinal differences in signal penetration depth between the September and winter SAR data.

When transferred to all areas on Novaya Zemlya, the glacier surface acquired by TanDEM-X in September 2016 was approximately 2.3 m higher than the surface elevations measured during the winter months 2016/17. The uncorrected glacier mass change rate of Novaya Zemlya is therefore ~20% lower than the corrected mass change because the elevation changes derived from DEM acquisitions of September 2016 is consistently less negative than those from the winter months.

Glacier surface elevation changes of the Russian Arctic Archipelagos are shown in Fig. 2. High thinning rates are measured at elevations below 600 m a.s.l., while surface change rates in the upper accumulation areas are close to zero or slightly positive. Average elevation change rates are highest on Novaya Zemlya ($\Delta h/\Delta t$ $_{corr.}$ = -0.64±0.46 m a$^{-1}$), mostly due to strong surface thinning close to the termini of the large outlet and tidewater glaciers (Fig. S3c). Regional average elevation changes of glaciers in Franz Josef Land (-0.48±0.04 m a$^{-1}$) and Severnaya Zemlya (-0.34±0.12 m a$^{-1}$) are in general less negative and strong thinning is confined to a smaller number of glaciers. Average elevation changes on Severnaya Zemlya are strongly positive below 50 m a.s.l. (Fig. S3b) due to a surge event within the observation period at the Vavilov Ice Cap (Zheng et al., 2019). Slight thickening is observed at the highest glacierized altitudes and the Academy of Sciences Ice Cap (Severnaya Zemlya), similar to the observations of (Sánchez-Gámez et al., 2019). The overall adjusted mass change of the Russian Arctic is -22.19±6.41 Gt a$^{-1}$ (density conversion factor: 850 kg m$^{-3}$). Approximately 50% of the total mass loss are caused by glaciers on Novaya Zemlya, while mass changes of Severnaya Zemlya and Franz Josef Land account for about a quarter each. Table 1 summarises the uncorrected and corrected change rates for the Russian Arctic.

## 4. Discussion

Differences in the SAR derived elevation change rates (Fig. 1b) can be related either to surface penetration of the X-Band radar or physical changes of the surface height due to accumulation or ablation of snow and ice. The TanDEM-X DEM difference on Novaya Zemlya does not fully cover the accumulation period of the last year of the observation period as the

acquisitions of September 2016 do not or only partially capture the amount of winter accumulation from October to December. This potential bias in measured winter accumulation would lead to an overestimation of surface elevation loss between winter 2010/11 and September 2016. However, the analysis of surface elevation changes derived from September and winter DEMs indicate, that the surface measured by TanDEM-X in winter 2016/17 was below the surface heights acquired in September 2016. As the occurrence of major surface melt within the Arctic winter months is unlikely, the observed elevation offset is most likely related to differences in the relative depth of signal penetration of the X-band SAR. The analysis of backscatter intensities of different acquisition months (Fig. 1a) indicates a change in glacier surface properties between the acquisitions from September 2016 and winter 2016/17. The observed differences in backscatter could be related either to the occurrence of melt or presence of fresh snow at the glacier surface which would decrease the depth of signal penetration as the amount of penetration depends on the condition of the glacier surface and is close to zero for melting snow surfaces and bare glacier ice. The majority of SAR data of the 2016/17 timestep was acquired at months with temperatures well below 0°C while average temperatures on Novaya Zemlya in September 2016 were close to the melting point (Fig. S1f). Thus, the differences in backscatter intensity could be caused by either accumulation of fresh snow at the glacier surface or days with snow melt during September 2016. It is likely that the depth of signal penetration in the winter seasons 2010/11 and 2016/17 was relatively large, i.e. several meters as found by previous studies (Millan et al., 2015; Zhao and Floricioiu, 2017; Abdullahi et al., 2018; Li et al., 2021), but similar due to comparable dry and frozen surface conditions. For TanDEM-X DEMs of the Antarctic Peninsula it was observed that the measured cold-season heights rather referred to the refrozen firn of the previous summer than to the actual glacier surface (Rott et al., 2014). This might be also the case for some of the glacierized areas of Novaya Zemlya where the observed bias between September and winter DEMs is relatively small (e.g. < 2 m). However, for glacier areas with higher differences in signal penetration depth, it is more likely that the measurement is biased by penetration beyond the previous late-summer surface, either by an older ice layer of a year with widespread melt and refreezing or volume scattering of the X-band SAR (Dall et al., 2001). Either way, during the September 2016 acquisitions on Novaya Zemlya, the absolute depth of signal penetration was probably smaller and the measured surface closer to the glacier surface. However, due to the change in surface conditions, the relative difference between penetration depths of the winter season 2010/11 and September 2016 increased. The corrected glacier elevation change rate of Novaya Zemlya is therefore more negative than the uncorrected rate because the effects of different signal penetration depths probably outweigh the winter accumulation. It is noteworthy, that the applied regional correction scheme can introduce a larger uncertainty at a local glacier scale caused by different surface and backscatter conditions between the specific TanDEM-X acquisitions (Fig. S2b). However, due to the limited extent of overlapping glacier areas (Fig. S2a), it is not possible to derive a date-specific intensity correction for each DEM strip. Thus, the applied linear model does rather represent an average difference in surface penetration depth between September and winter SAR data.

Over the last decades, the High Arctic has been subject to ongoing warming (Jansen et al., 2020) and glacier mass budgets have become more negative. Compared to previous studies (Fig. S5), glacier mass loss has increased in the Russian High Arctic since 2010. The glacier mass changes measured by TanDEM-X are similar or more negative than recent gravimetric

records (Ciracì et al., 2020; Wouters et al., 2019), supporting their observation of increasing mass loss. Recent large-scale regional studies based on optical elevation models (Hugonnet et al., 2021) and altimetry (Tepes et al., 2021) reported less negative mass changes (-10.4±1.9 Gt a$^{-1}$ and -14.0±0.5 Gt a$^{-1}$) since 2000, yet their measurements also indicate a distinct acceleration in mass loss over the course of the 21$^{st}$ century.

While the regional geodetic mass change derived from TanDEM-X data of Franz Josef Land is very similar to recent gravi-metric (Ciracì et al., 2020) and altimetric (Zheng et al., 2018) measurements, the estimate for Severnaya Zemlya is even more negative than the gravimetric measurements of (Ciracì et al., 2020) which might indicate recent acceleration of glacier mass loss also on this archipelago. However, the highest mass changes are mostly confined to a small number of outlet glaciers of the Vavilov and Academy of Sciences Ice Caps (RGI60-09.00915,919,920,971). For the remaining glacierized areas of the Severnaya Zemlya Archipelago (~14,000 km²), the mass change rate is much smaller (-2.39 Gt a$^{-1}$, 850 kg m$^{-3}$) than for the entire region (-4.70 Gt a$^{-1}$).

The strongest local surface lowering is observed at some of the large marine-terminating outlet glaciers, most notably on Novaya Zemlya (Northwestern Severny Island ice cap). For those glaciers, an increasing retreat in the early 21$^{st}$ century was attributed to fjord geometries and changes in sea-ice concentrations (Carr et al., 2014). Long-term observations indicate a more rapid thinning during recent years, particularly at the termini of marine-terminating glaciers (Melkonian et al., 2016). An acceleration in flow velocities for the major tidewater glaciers in the Russian Arctic was also measured by (Strozzi et al., 2017) over the course of the last decades. Using a combination of gravimetric and altimetric measurements, (Ciracì et al., 2018) reported a similar mass change of -14±4 Gt a$^{-1}$ for Novaya Zemlya (2010-2016) to the corrected mass change rate derived by TanDEM-X.

In contrast to the lower ablation areas, elevation gains of up to 0.4 m a$^{-1}$ are measured for the highest altitudes of the Russian Arctic archipelagos, which do not seem to be related to differences in SAR penetration because the respective measurements were acquired under similar surface conditions. This is particularly noticeable at some parts of the large accumulation areas of Novaya Zemlya, Severnaya Zemlya and Graham Bell Island (Franz Josef Land). Similar patterns can be also observed in the elevation change maps of altimetry measurements (Ciracì et al., 2018; Moholdt et al., 2012; Sánchez-Gámez et al., 2019; Zheng et al., 2018) and might be related to increased moisture transport and accumulation (Box et al., 2019). The ERA5 datasets also indicate a positive trend in temperature and total column water vapor (Fig. S4a & b) for the Russian Arctic archipelagos. However, the latter trend is not statistically significant in most regions and less pronounced than the increase in temperature, supporting our observations of an overall amplification of glacier mass loss.

## 5. Conclusion

Glaciers in the Russian High Arctic have shown a contribution of 0.06 mm a$^{-1}$ to global sea-level rise between 2010 and 2017 and an increased mass loss compared to the first decade of the 21$^{st}$ century. This observation is in line with glacier changes of other Arctic regions, showing an increasing contribution to sea-level rise in the last decades. While specific mass change rates

of Arctic glaciers are still less negative than those of many glaciers outside the polar regions, the absolute mass loss is higher due to the vast glacierized areas of the Arctic.

The acquisition date related differences in elevation change on Novaya Zemlya highlight the relevance of similar surface
conditions between SAR acquisitions when using DEM-differencing. Particularly for shorter observation periods, corrections for temporal offsets between acquisitions are crucial as the uncorrected elevation change rate can be biased by changes in surface conditions. However, acquisitions from the same season should be used whenever possible, as the measurement uncertainty increases depending on the corrected glacier area. Regarding upcoming TanDEM-X acquisitions, combined measurements with the new ICESat-2 laser altimeter have the potential to much better constrain offsets between different acquisition
dates.

*Data availability.* Elevation change maps and raster masks with the specific observation period of each cell are provided via the World Data Center PANGAEA (https://www.pangaea.de/) at #link#.

*Author contributions.* C.S. processed the glacier elevation and mass change data, created the graphs and wrote the manuscript. The analysis code of the DEM creation and coregistration was jointly developed by C.S. and T.C.S. A.G. contributed to the
210 comparison and interpretation of measured glacier changes. M.H.B. initiated and led the study. All authors revised the paper.

*Competing interests.* The authors declare that they have no conflict of interest.

*Acknowledgements.* This study was financially supported by the grant BR2105/14-2 within the DFG Priority Program "Re-
215 gional Sea Level Change and Society". We thank the Copernicus Climate Change Service (C3S) which is implemented by the European Centre for Medium-range Weather Forecasts (ECMWF) on behalf of the European Commission for free and open data access. TanDEM-X data was kindly provided free of charge by the German Aerospace Center (DLR) under AO mabra_XTI_GLAC0264.

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

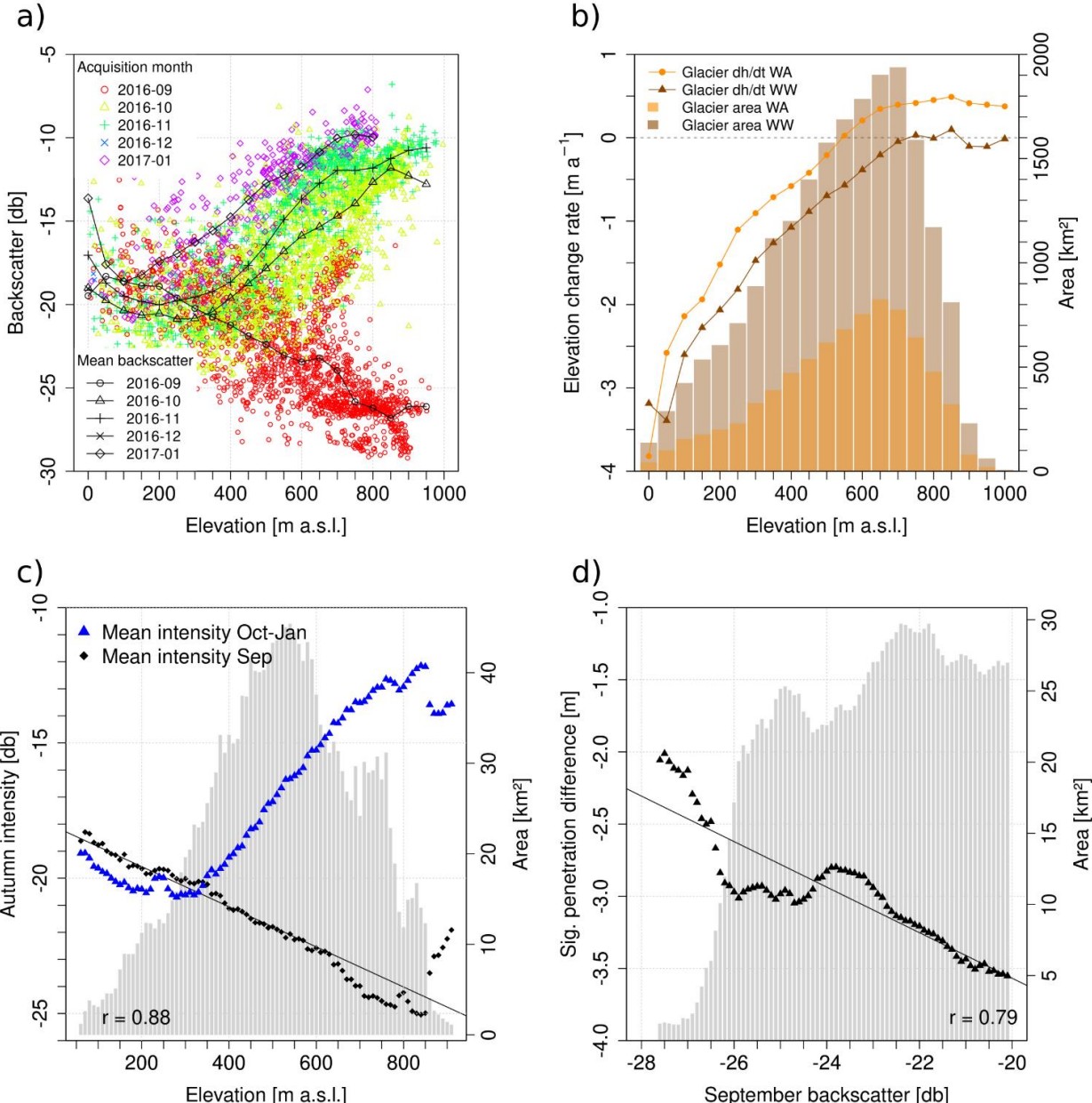

**Figure 1 a) Backscatter intensity of different TanDEM-X DEM acquisition months versus elevation on Novaya Zemlya. Black lines indicate average backscatter aggregated within 50m elevation bins. Point icons illustrate a random subset (5000 cells) of the 2016/17 DEM mosaic of Novaya Zemlya. In December 2016 (blue crosses), only a small glacier area at the Northeastern coast was acquired.**
**b) Mean elevation change rates of DEM differences between winter 2010/11 and winter 2016/17 (WW, triangles) and winter 2010/11 and September 2016 (WA, dots) of all respective glacier areas. c) Altitudinal distribution of mean backscatter intensity (aggregated in 10m elevation bins) of September and winter 2016/17 SAR data on overlapping glacier areas (i.e. areas which were acquired in September and winter 2016/17). d) Differences in estimated signal penetration between September and winter 2016/17 (supplement 3.1.) versus mean backscatter intensity of September 2016 acquisitions on overlapping glacier areas (aggregated within 0.1 db backscatter intervals between -20 to -28 db).**
**backscatter intervals between -20 to -28 db). The linear correlations of mean September backscatter intensity and elevation (1c) and mean difference in signal penetration depth and September backscatter intensity (1d) are indicated as black solid lines.**

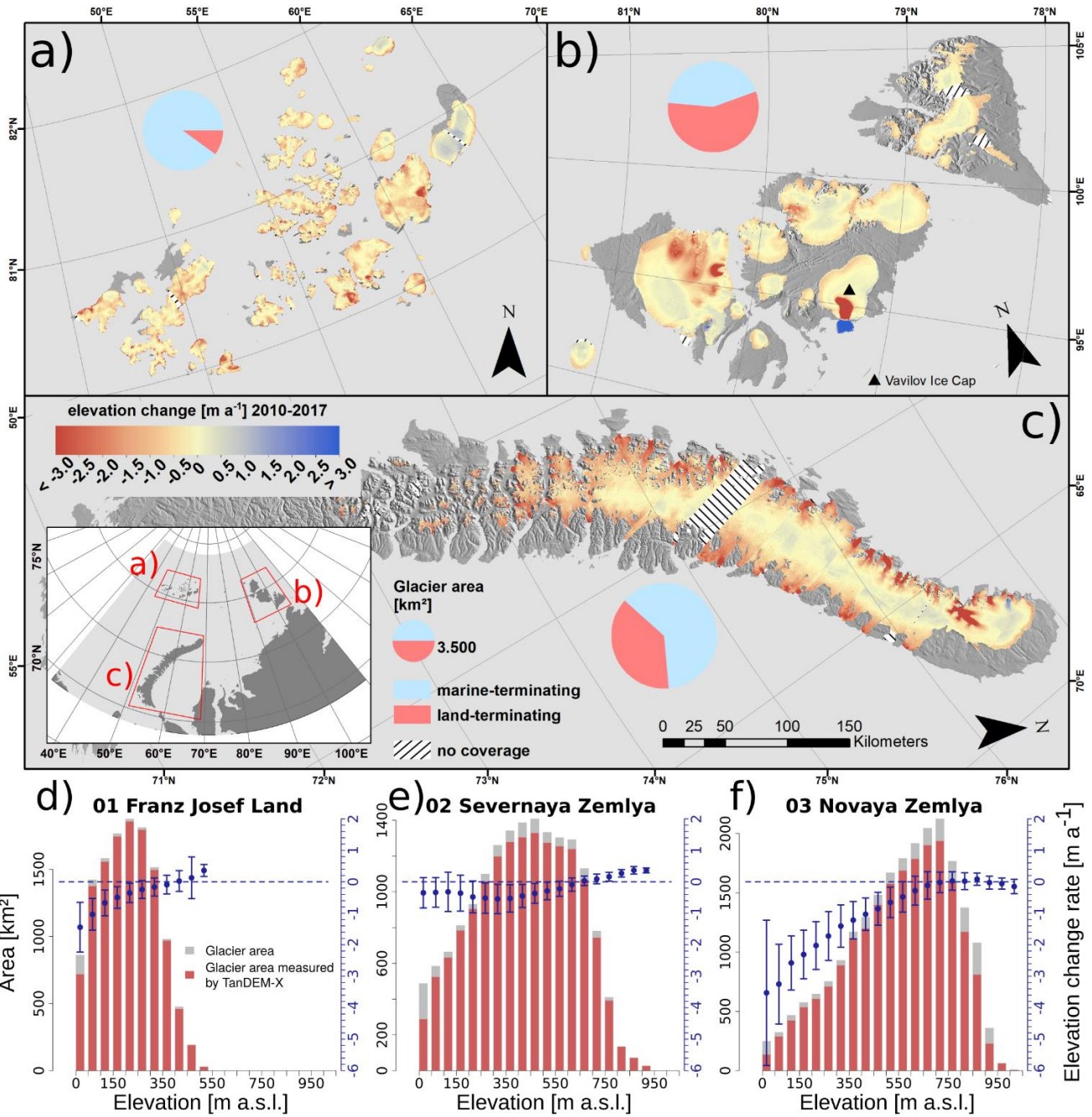

**Figure 2: Surface elevation changes of glaciers on Franz Josef Land (a), Severnaya Zemlya (b), and Novaya Zemlya (c) between 2010 and 2017. Hatched areas indicate glaciers without coverage by TanDEM-X. Respective average elevation change rates and total/measured glacier areas within 50m elevation bins are shown in Figures 1 (d)-(f). Blue vertical bars indicate the normalized median absolute deviation of elevation change measurements of each elevation bin. The hypsometric distribution of Severnaya Zemlya does not include the surge of the Vavilov ice cap (RGI60-09.00971). Elevation changes of Novaya Zemlya were corrected for differences in seasonal SAR-signal penetration (Fig. S2).**


**Table 1: Overview of glacier elevation and mass change in the Russian Arctic between 2010/11 and 2016/17. Glacier areas (*S*) are derived from the Randolph Glacier Inventory 6.0. Its spatial coverage by elevation change measurements (*S* mea.) is stated in percent. *Δh/Δt uncorr.* shows elevation change rates as measured by TanDEM-X while *Δh/Δt corr.* includes the SAR signal-penetration corrected elevation change rate of Novaya Zemlya. *ΔM/Δt uncorr.* and *ΔM/Δt corr.* are the respective glacier mass change rates using a volume-to-mass conversion factor of 850 kg m$^{-3}$ (a) and 900 kg m$^{-3}$ (b).**

**\*Acquisition date offsets corrected for Novaya Zemlya.**

| Region | S [km²] | S mea. [%] | Δh/Δt uncorr. [m a⁻¹] | Δh/Δt corr. [m a⁻¹] | ΔM/Δt uncorr. [Gt a⁻¹]a | ΔM/Δt corr. [Gt a⁻¹]a | ΔM/Δt uncorr. [Gt a⁻¹]b | ΔM/Δt corr. [Gt a⁻¹]b |
|---|---|---|---|---|---|---|---|---|
| Franz Josef Land | 12750 | 96 | -0.48±0.04 | | -5.14±0.43 | | -5.45±0.45 | |
| Severnaya Zemlya | 16529 | 97 | -0.34±0.12 | | -4.70±1.31 | | -4.98±1.38 | |
| Novaya Zemlya | 22117 | 91 | -0.53±0.23 | -0.64±0.46* | -9.95±3.14 | -12.06±6.17 | -10.54±3.31 | -12.76±6.53* |
| Russian Arctic | 51707 | 93 | -0.46±0.15 | -0.52±0.24* | -20.05±3.47 | -22.19±6.41 | -21.23±3.66 | -23.49±6.78* |

