# Peer review of "Brief communication: Increased glacier mass loss in the Russian High Arctic (2010-2017)"

_The Cryosphere, 2020_

## Referee Comment (RC1) · Anonymous Referee #1 · 28 Feb 2021

The comment was uploaded in the form of a supplement:
https://tc.copernicus.org/preprints/tc-2020-358/tc-2020-358-RC1-supplement.pdf

---

## Referee Comment (RC2) · Anonymous Referee #2 · 4 Mar 2021

This manuscript provides new geodetic estimates of glacier mass balance for the three main Russian Arctic archipelagos (Novaya Zemlya, Severnaya Zemlya, Franz Josef Land) and briefly discusses the results. The two most novel aspects of the study are that near-complete coverage of glacier elevation changes is obtained, and that the results indicate an increase in mass loss compared to earlier periods and studies.

The authors use digital elevation models (DEMs) derived from SAR interferometry of the TanDEM-X mission. This has the advantage of providing near-complete repeat coverage of glacier areas (93%), but can suffer from variable X-band radar signal penetration in snow/ice between satellite acquisitions. This is one of the main discussion points of the paper, and a correction-scheme is proposed for Novaya Zemlya where seasonal acquisition times were most different. Meteorological reanalysis data and

supplementary DEM analyses are presented to support the approach, and results are provided both with and without penetration correction, as well as for two different density assumptions in the conversion between volume and mass change.

The main results appear plausible and relatively robust overall, but the differences related to acquisition times on Novaya Zemlya are puzzling and do not give a strong justification for the applied correction scheme. The potential magnitude and mechanisms of seasonal penetration differences are not well described or discussed, and the relevant parts of the manuscript (mainly Section 2.3) brings more confusion than clarity. For example, the paper does not say anything about the spatial coverage of the autumn and winter data of 2016/17 (Do they cover areas of potential different glacier change? Is there any overlap so that the two periods can be compared directly?) or if winter snow is partly accounted for in the co-registration process over land areas, which would limit the need for seasonal correction. See the specific comments below for further details on this issue.

The manuscript is written in a Brief Communication format, which is probably related with the authors' previous publications with similar methodology in other glacier regions, but I think that the present version suffers from too short/unclear methodology and very limited discussions. I think a lot of this can be fixed with improved writing and referencing, and perhaps by moving parts or all of Sections 2.2 (uncertainty assessment) and 2.3 (Dem Acquisition date correction) to the Supplement as these two sections are not satisfactory in the present form (see specific comments below). Alternatively, the manuscript could be expanded to a normal paper by making more complete data/methods sections and expanding the discussion of observed glacier changes which is now very brief. In any case, some major revisions are needed regarding these aspects.

Specific comments and edits:

Title: Since parts of Siberia is often considered to be in the Russian Arctic and there are

areas with small mountain glaciers there, it would be more precise to say "Russian High Arctic" or "Russian Arctic archipelagos" in the title and elsewhere in the manuscript. Also, I think that "increased" is a more correct term than "accelerated" considering your results in relation to other studies.

L7: I assume you mean "atmospheric warming" or "surface warming", not the thermal state of the glaciers.

L15: This reference only considers one region. Please provide a few other similar refs or a more general one covering multiple regions. Russian Arctic

L21: Or more broadly: "...and various corrections related to surrounding oceans, surface hydrology and glacial isostatic adjustment (GIA)."

L30: What is the CoSSC tile product? Write out the acronym as a minimum.

L30: "Compared with..." – what do you actually mean? "Unlike..." or "Similar to..."

L34: Did you cross-check this coastline against the glacier inventory to make sure no glacier areas were excluded? Please specify in the text to make this clear.

L35: This relates to the sentence at L30. Please combine similar content at one place.

L36: Somewhat unclear. After a few reads I understand it as .... 2010/11 co-registered to Global DEM and mosaiced ... then 2016/17 co-registered to the 2010/11 mosaic to make a 2016/17 mosaic. Please clarify the text.

L37: I understand this as dividing by decimal numbers of years according to the dates of the source tiles. But that's confusing since you are differencing DEM mosaics. Does that mean you also made a mosaic layer of time differences? Or did you divide by an integer number of years (6) everywhere which would make more sense in a climatic mass balance perspective? Either approach could be justified, but this not discussed at all although it could have a significant impact on the results.

L39: Would be good to refer Fig. 1 here since the altitude dh/dt function is shown there.

L40: Isn't the inventory applied earlier than this, e.g. for the void filling? Also, the inventory is somewhat outdated, so what was done (or not) for glaciers that have undergone major changes such as the advancing Vavilov ice cap. The altitude-dependency of dh/dt in Fig S2 indicates that the Vavilov advance has been accounted for, whereas the less negative dh/dt of the lowermost altitudes of land-terminating glaciers in NZ indicate an impact from retreat which shouldn't influence overall mass rates (Gt/y), but could impact the area-specific rates (m/y). A brief discussion of these matters would be good to have somewhere in the manuscript. Note that there is a newer inventory for Novaya Zemlya (Rastner et al., 2017) which could be relevant for context or comparison.

L42-43: It's not the scenarios that change, but the firn pack. Rewrite sentence to make it clear what you actually mean here. Also, do you consider this issue to be within the error estimates you provide or as something that comes on top of that (i.e. not considered).

L44: Unclear and not strictly correct. It does include frontal melt/calving when that balances the ice outflux, but it does not include subaqueous glacier volume changes related to advance or retreat. This should be made clear, and also its potential relevance for the overall glacier mass balance and sea-level contribution, here or in the discussion.

L45: The uncertainty section is not understandable by itself and needs to be rewritten. There are parameters that are not fully explained, units are unclear, and it is hard to follow the logic unless a lot of time is spent with Table S1 and given references.

Eq. 1: Is this equation from previous work or is it unique for this study? It appears like mass rate uncertainty is a factor of the mass rate itself which does not make sense to me if the mass rate turn out to be near zero.

L56: Is $S_g$ ever larger than $S_{cor}$ here? If not, then it's confusing to include this equation. I understand it as you are calculating errors per region, not per glacier.

L60: How was this number found? Not clear from Section 2.3. It is also unclear if the approximate 2 m penetration difference (Spen) is applied only to the NZ autumn data or to all data in all regions which would make most sense.

L64-79: I like the comparative elevation differencing from winter 2010/11 to autumn (WA) and winter (WW) 2016/17, respectively, and I agree it might be the best way to try to account for errors related to signal penetration, but the logic is too simplified. Is it just melting or non-melting surface condition that is relevant? Widespread melting conditions are unlikely after mid-September, and ERA5 is too coarse to capture topographic temperature variations. In that context, I would consider differences in SAR backscatter to be relevant. And how deep can the X-band signal penetrate? There is no mention or references regarding that. For example, is the last summer-surface a dominant reflection horizon during winter or can it penetrate even deeper. In the latter case, the meteorological conditions of previous years might also matter.

L84: Fig. S2 shows altitude dependency, not whether a glacier is small or large. Rephrase or refer to Fig. 1 instead where it does seem like the largest glacier fronts thin the most.

L94: Unclear. Rather something like this: "Relations between acquisition times, monthly temperatures and derived elevation change rates for NZ are shown..."

L98: Redundant wording; elevation gains are always positive.

L102-104: True if no penetration, whereas if fresh cold snow is transparent then it can be considered as autumn 2010 to autumn 2016 changes, with no seasonal snow bias.

L112: This is also what I speculated (see previous comment), but then dh/dt from the WA and WW periods should have been more or less similar, which is not the case.

L113-115: I don't understand the logic here. Are you suggesting penetration into the firn/ice during winters and near-surface reflection during autumn? If so, you are in practice measuring a "delayed mass balance" (shifted backwards in time).

L117: The figure indicates largest warming for the northern islands (FJL and SZ) and smallest for the southern ones (NZ), which is opposite of what you say. But warming might still have a larger impact in the south since climate is in general warmer and closer to the melting point. The most relevant aspect for this paper would be how 2010-17 stands out from the longer-term climate, especially during the summer melt season. Any relevant references that have studied climate change in this region in more detail?

L119: What about the comparable Wouters et al. (2019) paper?

L121: How much of your mass loss is related to the surge of Vavilov ice cap? Would there be a substantial remaining mass loss if dynamic areas of Vavilov and Academy of Sciences ice caps were excluded? I miss such aspects of the discussion.

L123-l30: The study of Melkonian et al. (2016) is also very relevant for this discussion, considering both long-term elevation changes and ice dynamics.

L129: are not always related to -> does not seem to be related to

L132: Zhang et al. (2018) is also very relevant here (only referenced in the Supplement)

L137: showed -> has shown

L138: ...between 2010 and 2017

L139: Unclear. Do you mean that Arctic glacier mass losses are increasing more than non-polar ones? If so, in total or specific rates?

L140: You are basically listing all regions except Svalbard. Is this sentence needed?

Fig. 1: Nice figure. Is it possible to also show the autumn (A) versus winter (W) coverage of DEMs in the 2016/17 seasons? Or in the supplement to keep this figure clean.

[Figure]

Table S1: You seem to use AW here as an abbreviation for area-weighted, which is confusing because you use AW as an abbreviation for autumn-winter elsewhere in the manuscript. And at L51 you write slope-weighted instead of area-weighted.

Fig. S1: Are the climatological data extracted for the entire regions or specifically for the glacier areas? I don't think that is mentioned anywhere in the manuscript.

Fig. S4: Nice compilation of results. For FJZ, itt should be Zheng et al. (2018), not 2019 which is another paper.

References

Melkonian, A. K., M. J. Willis, M. E. Pritchard, and A. J. Stewart (2016), Recent changes in glacier velocities and thinning at Novaya Zemlya, Remote Sens. Environ., 174, 244-257.

Rastner, P., T. Strozzi, and F. Paul (2017), Fusion of Multi-Source Satellite Data and DEMs to Create a New Glacier Inventory for Novaya Zemlya, Remote Sensing, 9(11).

---

## Author Comment (AC1) · 11 Apr 2021

**Paper**
**Brief communication: Accelerated glacier mass loss in the Russian Arctic (2010-2017)**

The Cryosphere Discuss. https://tc.copernicus.org/preprints/tc-2020-358/
**Comments to the authors**

**1. Summary and general comments**

The presented work estimates the mass balance of three glaciated archipelagos of the Arctic Ocean in Northern Russia, namely Novaya Zemlya (NZ), Severnaya Zemlya (SZ) and Franz Josef Land (FJL). The three groups of islands are largely glaciated and were subject of several investigations related to their ice mass loss in the recent years using gravimetry and altimetry data. This study is based on elevation data from bistatic SAR satellite mission TanDEM-X and applies the meanwhile well-established method of calculating the ice surface elevation difference between DEMs acquired during the mission at different dates (here the winters 2010/2011 and 2016/2017). The methodology is one of the most precise for estimating spatially distributed, high resolution surface elevation change rates.

However, since NZ's mass loss is the largest of the three archipelagos (50% of the total) and because 70% of the TanDEM-X data in the winter 2016/2017 mosaic were acquired earlier, namely in September/October 2016, while the other two smaller archipelagos have each about a quarter contribution to the mass loss and the TanDEM-X data processed here were acquired in the same season a particular attention has to be given to the processing and analysis of NZ. Two problems arise here regarding the measured elevation changes:

1. the glaciological cycle is not fully covered missing parts of the accumulation period in the elevation change rate dh/dt.
2. the different reference surfaces of the InSAR DEMs from late summer/early autumn vs. winter introduce apparent changes in surface elevation due to differences in radar signal penetration.

Although these issues are addressed along the paper they are not clearly separated and the effects on the results are confusing. The uncertainty assessment of the mass change rate presented in this paper consists of three terms, the vertical coregistration being explained in detail. This term includes also the effect of SAR signal penetration as the factor $S_{pen}$ (eq 2) resulted from the winter-autumn (WA) and winter-winter (WW) elevation changes of NZ. The authors apply a bulk correction which is a questionable approach because the surfaces of NZ glaciers extend over an elevation range of 1500 m and thus include ice/snow volumes of very different penetration properties, from close to zero for glaciers ice to several meters in the upper sections of the accumulation area (if dry). The elevation range implies a typical temperature difference of about 10 °C, so that – in particular in late summer and autumn – surface melt all over is rather unlikely. This indicates the need for localised penetration corrections, e.g. using backscatter coefficients in SAR amplitude images for assessing the melting state. The SAR backscattering coefficient on each archipelago is a more precise indicator than one mean monthly value of skin temperature (see specific comments).

Initially we would like to thank the reviewer for the detailed and comprehensive comments. We agree that the correction of elevation differences (likely related to signal penetration) based on monthly average temperatures is a coarse correction approach and replaced it with a more detailed analysis of backscatter intensity on overlapping glacier areas which were acquired in autumn and winter 2016/17. Therefore, we moved the description of the interferometric DEM creation and co-registration to the supplement because those sections closely follow previous publications. Instead, we extended the description and discussion of radar signal penetration in the main manuscript (new chapter 2.2 and 2.3) and included an analysis of backscatter intensities of different TanDEM-X

**acquisitions on overlapping glacier areas of Novaya Zemlya (i.e. glacier areas which were acquired during autumn and winter 2016/17). Thereafter, we apply an intensity based correction for the remaining autumn 2016 DEMs. We also included further figures (e.g. spatial distribution of different DEM acquisition dates).**

2. **Specific comments, minor comments & typos**

Line 55: …× $\sigma^2_{\Delta h/\Delta t\,AW}$… (in case this equation comes from the spherical variogram model)
**\*Equation corrected**

Line 57: correct subscripts (Scor, SG)
**\*Subscripts corrected**

Line 73: delete **/17** after autumn 2016. The explanations of the polynomial correction are insufficient and the results over NZ aren't traceable. See comments below.
**\*We changed the correction estimate for Novaya Zemlya and extended the respective methods section.**

Line 75 ff. melting/ penetration: the presence of melt should be assessed by checking the backscatter coefficients. In case differences in sigma0 between the data used for retrieving the elevation change are indicating differences in signal penetration, these should be corrected.
**\*Included analysis of backscatter intensity and respective correction**

Lines 77 and 78: replace "images" with "TanDEM-X data" or "SAR data"
**\*Replaced "images"**

Line 94 to 98 and Fig. 2a: According to this analysis the period December-April 2010/11 to November-January 2016/17 (WW) shows higher average rates of elevation loss (dh/dt) than the period December-April 2010/11 to September-October 2016 (WA). This is contrary to the expected behaviour if the annual mass balance cycle is taken into account (as I mentioned at point 1. the winter accumulation is partly missing). A possible explanation could be a bias in the penetration correction. Fig. 2b: Novaya Zemlya extends from 71 N to 77 N and 0 m to 1500 m a.s.l. A single mean monthly mean skin temperature is not a useful indicator for estimating the melting state as major spatial and temporal differences have to be expected.
**\*The discussion on potential effects of "missing" winter accumulation and differences in signal penetration was extended. The temperature-based estimate for Novaya Zemlya was replaced by an analysis of backscatter intensity (new chapter 2.2 & 2.3).**

Line 100 ff.: Which "differences in the SAR derived elevation change rates" exactly? Cross reference Fig 2a if you are referring to the dh/dt of NZ for WW and WA periods. Or is a general statement?
**\*Yes, this refers to autumn/winter DEMs on NZ. Included reference to Fig 2a**

Line 104: If accumulation is only partly included in the 2016/17 elevation this would lead (without applying a correction) to an overestimation of the surface elevation loss.
**\*We replaced "overestimation of surface elevation change" with "overestimation of surface elevation loss" to clarify this sentence.**

Lines 105: … elevation change of the WA period is less negative than of the WW period at all altitudes, …
**\*Corrected and rephrased sentence.**

Line 107 replace "decreases during melting conditions" with "is close to zero for melting snow surfaces and for bare glacier ice in general".
**\*Ok**

Line 117: I see in Fig S3a reddish areas (warmer temperatures) on the ocean and on the northern islands (FJL, SZ) not on the southern islands (NZ). The skin temperature does not show any temporal trend for the glaciers on NZ, supporting the comment above that the temperatures shown in Fig. 2b are not representative for the main glacier areas. The skin temperature increase is most pronounced on the ocean, due to the decrease in sea ice coverage.

*We agree that the warming trend in skin temperature is most pronounced on the northern archipelagos and ocean areas. However, the ERA5 dataset indicates a general increase in temperature for the entire region and the interior of Novaya Zemlya (<= +1 C°/dec). In Fig. S1 (monthly mean temperatures of TanDEM-X acquisition months) we also replaced the ERA5 data with the ERA5 Land dataset which provides a better spatial resolution and does not include ocean areas.

Line 133: "indicate"
*Ok

Line 254: Caption of Table 1: Overview of glacier elevation and mass change in the Russian Arctic between 2016 and 2017. This is probably a typo and should mean "2010/2011 to 2016/2017".
*Yes, replaced wrong year numbers

Line 275: The **a)** and **b)** notations on Fig 2 are missing.
*Extended figure and added missing notations.

**Supplement**

Fig. S1: Total precipitation units cannot be meter/day, but mm/day. Please add reference/data source for the ERA5 temperatures and precipitation and give the locations of the measurements.
*We changed Fig. S1 and removed the precipitation plots. For the temperature values, we added the respective reference and locations of the used reanalysis data in the caption and in the figure.

Fig S3 the **a)** and **b)** notations are missing. But references to climate data related studies of the same period are more convincing in demonstrating the long term trends than the trends shown in these figures.
*We added the notations to Fig S4 (former Fig S3) and included a reference to a recent study about Arctic climate change in the discussion section of the main manuscript.

Please add a table with the specifications for the TanDEM-X database used in the study.

*We added Table S2 to the supplement which includes metadata of the used TanDEM-X acquisitions.

---

## Author Response (AR1)

**Author´s response to reviews of:**

**Brief communication: Increased glacier mass loss in the Russian High Arctic (2010-2017)**

Christian Sommer[1], Thorsten Seehaus[1], Andrey Glazovsky[2], Matthias H. Braun[1]

[1]Institut für Geographie, Friedrich-Alexander-Universität Erlangen-Nürnberg, Erlangen, 91058, Germany
[2]Institute of Geography RAS, Moscow, 119017, Russia

*Correspondence to*: Christian Sommer (chris.sommer@fau.de)

Initially, we would like to thank the reviewers and editor for the detailed and comprehensive comments on our manuscript "Increased glacier mass loss in the Russian High Arctic (2010-2017)". Following the suggestions during the review process, we changed the structure of the paper in the revised version. We moved parts of the methods section (DEM creation & co-registration) to the supplement as this approach has been already described in a number of previous publications. Instead, we extended the discussion and correction of radar signal penetration in the main manuscript. Therefore, we included an analysis of backscatter intensity, as suggested by the reviewers, and estimated the relative difference in surface penetration depth of different TanDEM-X acquisitions based on an empirical relationship between autumn backscatter and observed differences in measured surface elevation. All relevant changes made in the manuscript are summarized on page 2.

**Table of Contents**

**1) List of changes**

a) Methods section (chapter 2)
- Moved former chapter 2.1 (DEM creation) & 2.2 (uncertainty analysis) to supplement and extended method description
- Included short methods overview (2.1) and signal penetration properties (2.2) in main manuscript L 29-51
- Rewrote chapter 2.3 and included backscatter analysis L53-90

b) Results section (chapter 3)
- Included observations from backscatter analysis in results section L 104-116

c) Discussion section (chapter 4)
- Rephrased discussion of temporal differences between acquisitions on Novaya Zemlya L 118-141
- Slightly extended discussion and comparison between TanDEM-X derived mass change and previous studies L 142-168

d) Figures (main manuscript)
- Former Figure 1 (elevation change map) was moved to Figure 2
- Included new Figure 1 with results from backscatter analysis of Novaya Zemlya

e) Supplement
- Included extended methods section on datasets, DEM-creation & co-registration (p. 2-4)
- New Table S1 with DEM co-registration statistics on non-glacierized areas (p. 5)
- New Figure S1 with backscatter intensities of each subregion (p. 6)
- New Figure S2 with overview map of DEM acquisitions and surface penetration correction of Novaya Zemlya (p. 7)
- Included Table S2 with metadata of used TanDEM-X data (p. 11-18)

**2) **Response to the editor**

Comments to the Author:

ORIGINALITY / NOVELTY

The mass loss of glaciers in the Russian Arctic has already been estimated using various techniques but this is the first time, to my knowledge, this is done using DEM differencing. This study provides a welcome, novel estimates that helps to constrain the acceleration of the mass loss in this high arctic region.

SCIENTIFIC QUALITY / RIGOR

The methods used here have already been applied and validated elsewhere. I missed a clearer explanation of the seasonal corrections (see below). Also the density conversion value should be better justified.

SIGNIFICANCE / IMPACT

This is a brief and solid study. To increased its impact, the comparison to earlier studies could be improved (same time periods see below) and receive more space in the main text.

PRESENTATION QUALITY

The material is concisely presented. The text is also well-written.

**R: Thank you very much for the quick review and constructive comments. Our point-by-point responses are listed below:**

Acquisition date correction. It seems to me that two effects need to be distinguished, correct me if I am wrong. First, the fact that measurements are not made at the same of the year and thus, because of the strong elevation/mass balance seasonal cycle, this may lead to overestimation of mass loss (because the changes are computed from an annual elevation maximum in winter to a minimum in Autumn). The second effect is radar penetration. I think it would help/convince the reader/reviewers if these two effects were well separated.

**R: The differences in elevation change between the 2016/17 acquisition dates could be in fact related to either "real" physical changes of the surface heights (winter accumulation) or varying depths of radar signal penetration. The largest temporal offset exists for DEMs which were acquired in September 2016 (~35 % of total glacier area on Novaya Zemlya). For those areas, the accumulation of approximately 3 months of winter 2016/17 is not included in the observation period which would cause an overestimation of surface elevation change. Therefore, we included an analysis on surface elevation heights on overlapping glacier areas (i.e. areas which were measured in autumn and winter 2016/17 as well) in the revised manuscript. This comparison (Fig 1b) showed that actually the surface heights measured in September 2016 were at all altitudes higher than those measured in winter 2016/17. This is contradictory to the assumption of "missing" winter accumulation for the period winter 2010/11 to autumn 2016/17 because the presence of widespread melt in the High Arctic after September is very unlikely. Based on this observation, it is likely that the differences are related to varying depths of surface signal penetration between autumn and winter 2016/17.**

**Thus, we included an analysis on backscatter intensity, as suggested by both reviewers, which can be used to estimate signal penetration depths (e.g. Abdullahi et al., 2019). Using the backscatter**

**intensity and observed differences in measured surface heights (on all overlapping areas), we applied an empirical relation between September backscatter intensity and relative difference in surface penetration depth to adjust the elevation change rate of all glacier areas on Novaya Zemlya which were measured in September 2016.**

**To include the backscatter analysis and extended discussion on surface penetration, the former short methods description of the TanDEM-X DEM creation and co-registration was moved (and extended) to the supplement. This was also suggested by one of the reviewers. The new methods section in the main manuscript focuses on the observed differences in measured surface heights and backscatter. Additionally, we rephrased parts of the discussion section (see list of changes) to specifically discuss the observed relation between surface heights and radar backscatter properties.**

Comparison to earlier studies. GRACE studies provide a continuous time series. Contacting the first authors of recent studies (Wouters/Ciraci), authors could get their time series and extract the exact same time period as them and make a more convincing comparison. I looked at the Wouters et al. time series (Their Figure 2) and, visually, did not find an obvious signal of acceleration.

**R: An acceleration of mass loss in the Russian Artic ($-1.2$ Gt $a^{-1}$) over the period 2002-2019 was reported within the recent gravimetric study by (Ciracì et al., 2020). Also, for Novaya Zemlya, a gravimetric mass change measurement (Ciracì et al., 2018) for 2010-2016 was reported which is almost the same observation period as covered by TanDEM-X. We cited the respective publications in the comparison section of the discussion. However, we did not attempt to reconstruct the respective gravimetric mass change for the period 2010-2017 for Franz Josef Land and Severnaya Zemlya because it would be difficult to include this comparison within the page limits of the "Brief Communications" format in addition to the extended analysis on signal penetration.**

The 900 kg/m3 density need to be justified. It implies that the authors entirely neglect firn compaction. An assumption not so straightforward in the context of rapid warming of high latitude ice caps.

**R: We applied a volume-to-mass conversion factor of 900 kg/m3 in the initial submission to enable a straightforward comparison to some of the existing (gravimetric) regional studies which were conducted for the Russian Arctic archipelagos (e.g. Ciracì et al., 2018; Moholdt et al., 2012; Sánchez-Gámez et al., 2019). In the revised version we included additional mass change results using an average density scenario of 850 kg/m3 (Huss, 2013) and included an indication (Supplement Methods, L 64-65) that we do not consider potential changes in the density scenarios as there are no respective values available for this region.**

Figure 1. color blind people may not be able to distinguish the "no coverage" color from the (rare) positive dh/dt values.

**R: We changed the color scheme for the "no coverage" areas.**

Maybe for Severnaya Zemlya (panel e) it would be good to separate the signal from the surging glacier in Vavilov from the rest of the region. So that Severnaya Zemlya can be compared to the other regions. Otherwise the signature of the Vavilov collapse is really strong in the dh/dt curve with elevation.

**R: We removed the surging glacier part of Vavilov ice cap (RGI60-09.00971) from the hypsometric distribution of elevation changes of Severnaya Zemlya and adjusted the figure (and caption) accordingly.**

Table 1. For Franz Josef land and Severnaya (and not Severnaja as mis-written), I do not understand why the error bars are changing between the columns. Can the authors clarify this?

R: The error bars between the dh/dt columns are different as we intended to provide different measures of the vertical uncertainty of the provided dh/dt. The numbers of the 1st column represented the "raw" vertical offset on all areas outside glacier areas (slope-weighted standard deviations, $\sigma_{\Delta h/\Delta t\ AW}$) while the 2nd column showed the "final" dh/dt uncertainty ($\delta_{\Delta h/\Delta t}$) as described in the methods section (e.g. including interpolation, spatial auto correlation, …). To avoid confusion, we moved the uncertainty values of dh/dt column 1 from Table 1 to a new table with off-ice accuracy statistics in the supplement (Table S1).

Figure 2. The rational for separating marine/land terminating glaciers for this section (penetration effect) was not clear to me. Can the authors also clarify this point in the text? Why such a separation is needed.

R: In the original version we separated between marine- and land-terminating glaciers as we assumed potential differences in the dynamics of those glacier types (e.g. higher flow velocities of marine-term. Glaciers). However, the magnitude of signal penetration is in fact independent from the terminus type of the respective glacier. In the revised version we estimated differences in surface signal penetration by using observed elevation differences on overlapping glacier areas (i.e. areas which were measured in autumn and winter 2016/17). For this analysis we did not separate marine- and land-terminating glaciers.

**3)** **Response to reviewer 1**

**Paper**
**Brief communication: Accelerated glacier mass loss in the Russian Arctic (2010-2017)**

The Cryosphere Discuss. https://tc.copernicus.org/preprints/tc-2020-358/
**Comments to the authors**

**1) Summary and general comments**

The presented work estimates the mass balance of three glaciated archipelagos of the Arctic Ocean in Northern Russia, namely Novaya Zemlya (NZ), Severnaya Zemlya (SZ) and Franz Josef Land (FJL). The three groups of islands are largely glaciated and were subject of several investigations related to their ice mass loss in the recent years using gravimetry and altimetry data. This study is based on elevation data from bistatic SAR satellite mission TanDEM-X and applies the meanwhile well-established method of calculating the ice surface elevation difference between DEMs acquired during the mission at different dates (here the winters 2010/2011 and 2016/2017). The methodology is one of the most precise for estimating spatially distributed, high resolution surface elevation change rates.
However, since NZ's mass loss is the largest of the three archipelagos (50% of the total) and because 70% of the TanDEM-X data in the winter 2016/2017 mosaic were acquired earlier, namely in September/October 2016, while the other two smaller archipelagos have each about a quarter contribution to the mass loss and the TanDEM-X data processed here were acquired in the same season a particular attention has to be given to the processing and analysis of NZ. Two problems arise here regarding the measured elevation changes:

1. the glaciological cycle is not fully covered missing parts of the accumulation period in the elevation change rate dh/dt.
2. the different reference surfaces of the InSAR DEMs from late summer/early autumn vs. winter introduce apparent changes in surface elevation due to differences in radar signal penetration.

Although these issues are addressed along the paper they are not clearly separated and the effects on the results are confusing. The uncertainty assessment of the mass change rate presented in this paper consists of three terms, the vertical coregistration being explained in detail. This term includes also the effect of SAR signal penetration as the factor $S_{pen}$ (eq 2) resulted from the winter-autumn (WA) and winter-winter (WW) elevation changes of NZ. The authors apply a bulk correction which is a questionable approach because the surfaces of NZ glaciers extend over an elevation range of 1500 m and thus include ice/snow volumes of very different penetration properties, from close to zero for glaciers ice to several meters in the upper sections of the accumulation area (if dry). The elevation range implies a typical temperature difference of about 10 °C, so that – in particular in late summer and autumn – surface melt all over is rather unlikely. This indicates the need for localised penetration corrections, e.g. using backscatter coefficients in SAR amplitude images for assessing the melting state. The SAR backscattering coefficient on each archipelago is a more precise indicator than one mean monthly value of skin temperature (see specific comments).

**R: Initially we would like to thank the reviewer for the detailed and comprehensive comments. We agree that the correction of elevation differences (likely related to signal penetration) based on monthly average temperatures cannot fully explain local variations in surface conditions. Therefore, we replaced it in the revised version of the manuscript by a more detailed analysis of backscatter intensity and measured elevation change on glacier areas acquired in different months.**

**Concerning the main comments regarding the elevation change measurement on Novaya Zemlya:**

Author´s response: Increased glacier mass loss in the Russian High Arctic (2010-2017)

1) **As mentioned by the reviewer, the accumulation of Novaya Zemlya of the last year of the observation period is not entirely covered due to a temporal offset between the TanDEM-X acquisitions of 2010/11 and 2016/17. While the observation periods of most pixels start in winter 2010/11 (December & January), for some pixels the end date is in autumn 2016. Due to this shift, the observation period does not cover 2-3 months of winter accumulation for some glacier areas. Therefore, it is expected that the derived elevation change measurements would overestimate the actual surface lowering. Yet, the comparison of winter-to-winter and winter-to-autumn elevation change measurements indicate that the surface heights measured by TanDEM-X during winter 2016/17 were lower than the surface measured in autumn 2016 at all altitudes. It is not likely that this offset is primarily caused by physical changes of the surface heights between the acquisitions because widespread surface melt at all altitudes after late summer/autumn is not very likely in the Arctic. Also, the average vertical offset (> 2 m) between autumn and winter elevations (chapter 2.3, 3 & 4) is high compared to the average elevation change rate over the entire observation period. An analysis of backscatter (see point 2) on glacier areas, which were acquired in both seasons, indicates that the observed differences are rather related to differences in signal penetration depth (e.g. by days with melt in September) than to physical changes of the surface heights. We therefore provided a correction estimate which is based on the local backscatter intensity (see comments below).**

2) **We extracted backscatter intensities of all DEM data of Novaya Zemlya (2016/17) to analyze potential differences in surface conditions (and thereby differences in signal penetration) as suggested. By comparing the change of backscatter intensity versus elevation for different acquisition dates (Fig. 1), the acquisitions of September 2016 showed significant differences while the backscatter values of the September-January (2016/17) DEMs are relatively similar at all altitudes. The respective backscatter values are also displayed in Fig. 1a and Fig. S1 in the manuscript and supplement. While September backscatter intensity is relatively similar for altitudes below ~400 m a.s.l. and intensity values ~ -20 db, it diverges at higher altitudes. Thus, we applied the revised offset correction only for glacier areas which were acquired in September 2016 because the backscatter indicates the largest change in surface conditions for those areas.**

[Figure]

**Fig. 1 Backscatter intensity versus elevation of TanDEM-X acquisitions on Novaya Zemlya. Black lines indicate the mean DEM backscatter of each acquisition month.**

To account for the observed differences in backscatter between the acquisition dates we also revised the correction approach: We removed the bulk estimate of the mean elevation difference between September and winter DEMs. Instead, we applied a regression based on local backscatter intensity and altitude. To fit the correction model, we extracted the elevation difference and backscatter on all overlapping glacier areas (i.e. areas which were acquired in autumn and winter 2016/17). The mean backscatter of the overlapping areas diverges at altitudes above 400 m a.s.l. and backscatter intensities of approximately -20 db, which is similar as the altitudinal distribution of backscatter intensity of all acquisitions on Novaya Zemlya (Fig. 1). We then transferred the model to all glacier areas which were acquired only in September 2016 (Fig. 2a) and used the respective backscatter values to estimate a vertical correction value for each pixel (Fig. 2b). Fig. 2a (Fig. S2a in the supplement) shows the reference (overlap) areas as well as those areas which were eventually corrected. The vertical correction values are provided in Fig. 2b (Fig. S2b).

The analysis of the overlapping glacier areas and the transfer of the correction model to all other autumn DEMs is described in chapter 2.3 & 3 in the revised manuscript.

[Figure]

**Fig. 2 a) Overview of glacier areas which were acquired during September and winter 2016/17 (red dots) and areas which were measured only in September 2016 (blue triangles). b) Estimated vertical offset between September 2016 and winter 2016/17 TanDEM-X acquisitions.**

**Eventually, we made some changes of the manuscript structure to focus on the adjustments applied to the elevation change rate of Novaya Zemlya:**

**\*The methods sections describing the interferometric DEM creation and co-registration were moved (and extended) to the supplement (pages 2-4) because those sections closely follow previous publications.**
**\*The description and discussion of the penetration and elevation change on Novaya Zemlya in the main manuscript was extended.**
**\*Chapter 2.2 and 2.3 now describe the analysis and correction of different backscatter.**
**\*We changed the order of figures and extended Fig. 1 (former Fig. 2) with two additional panels on backscatter and estimated difference in signal penetration.**

**Please find our point-by-point responses below:**

**Specific comments, minor comments & typos**

Line 55: …× $\sigma^2_{\Delta h/\Delta t\ AW}$… (in case this equation comes from the spherical variogram model)
**R: Equation corrected**

Line 57: correct subscripts (Scor, SG)
**R: Subscripts corrected**

Line 73: delete **/17** after autumn 2016. The explanations of the polynomial correction are insufficient and the results over NZ aren't traceable. See comments below.
**R: We changed the correction estimate for Novaya Zemlya and extended the respective methods section. The original polynomial correction was replaced by a linear regression based on backscatter intensity and altitude (see general responses above and chapter 2.3, Line 71-83).**

Line 75 ff. melting/ penetration: the presence of melt should be assessed by checking the backscatter coefficients. In case differences in sigma0 between the data used for retrieving the elevation change are indicating differences in signal penetration, these should be corrected.
**R: Included analysis of backscatter intensity and respective correction (chapter 2.2 & 2.3). The DEM acquisitions of September 2016 showed large differences in backscatter and were accordingly adjusted.**

Lines 77 and 78: replace "images" with "TanDEM-X data" or "SAR data"
**R: Replaced "images"**

Line 94 to 98 and Fig. 2a: According to this analysis the period December-April 2010/11 to November-January 2016/17 (WW) shows higher average rates of elevation loss (dh/dt) than the period December-April 2010/11 to September-October 2016 (WA). This is contrary to the expected behaviour if the annual mass balance cycle is taken into account (as I mentioned at point 1. the winter accumulation is partly missing). A possible explanation could be a bias in the penetration correction. Fig. 2b: Novaya Zemlya extends from 71 N to 77 N and 0 m to 1500 m a.s.l. A single mean monthly mean skin temperature is not a useful indicator for estimating the melting state as major spatial and temporal differences have to be expected.
**R: As suggested, we replaced the mean monthly skintemperature by the backscatter intensity as indicator for changing surface conditions. The revised correction method uses local backscatter and altitude. This approach accounts for different altitudes and different states of the glacier surface across the Novaya Zemlya ice cap (see comments above and revised manuscript).**
**In addition, the discussion on potential effects of missing winter accumulation and differences in signal penetration was extended (chapter 4, Line 115-138).**

Line 100 ff.: Which "differences in the SAR derived elevation change rates" exactly? Cross reference Fig 2a if you are referring to the dh/dt of NZ for WW and WA periods. Or is a general statement?

**R: Yes, this refers to autumn/winter DEMs on NZ. Included reference to Fig 2a**

Line 104: If accumulation is only partly included in the 2016/17 elevation this would lead (without applying a correction) to an overestimation of the surface elevation loss.
**R: We replaced "overestimation of surface elevation change" with "overestimation of surface elevation loss" to clarify this sentence.**

Lines 105: … elevation change of the WA period is less negative than of the WW period at all altitudes, …
**R: Corrected and rephrased sentence.**

Line 107 replace "decreases during melting conditions" with "is close to zero for melting snow surfaces and for bare glacier ice in general".
**R: Ok**

Line 117: I see in Fig S3a reddish areas (warmer temperatures) on the ocean and on the northern islands (FJL, SZ) not on the southern islands (NZ). The skin temperature does not show any temporal trend for the glaciers on NZ, supporting the comment above that the temperatures shown in Fig. 2b are not representative for the main glacier areas. The skin temperature increase is most pronounced on the ocean, due to the decrease in sea ice coverage.
**R: We agree that the warming trend in skin temperature is most pronounced on the northern archipelagos and ocean areas. However, the ERA5 dataset indicates a general increase in temperature for the entire region and the interior of Novaya Zemlya (<= +1 C°/dec). In Fig. S1 (monthly mean temperatures of TanDEM-X acquisition months) we also replaced the ERA5 data with the ERA5 Land dataset which provides a better spatial resolution and does not include ocean areas.**

Line 133: "indicate"
**R: Ok**

Line 254: Caption of Table 1: Overview of glacier elevation and mass change in the Russian Arctic between 2016 and 2017. This is probably a typo and should mean "2010/2011 to 2016/2017".
**R: Yes, replaced wrong year numbers**

Line 275: The **a)** and **b)** notations on Fig 2 are missing.
**R: Extended figure and added missing notations.**

**Supplement**

Fig. S1: Total precipitation units cannot be meter/day, but mm/day. Please add reference/data source for the ERA5 temperatures and precipitation and give the locations of the measurements.
**R: We changed Fig. S1 and removed the precipitation plots. Instead backscatter intensity and average temperatures are compared for the acquisition months. Concerning the temperature values, we added the respective reference and locations of the used reanalysis data in the caption and in the figure. We also changed the reanalysis data and used the ERA5 Land product for the revised version instead of the ERA5 product. The ERA5 Land product does not include ocean areas and provides a better spatial resolution for land areas.**

Fig S3 the **a)** and **b)** notations are missing. But references to climate data related studies of the same period are more convincing in demonstrating the long term trends than the trends shown in these figures.
**R: We added the notations to Fig S4 (former Fig S3) and included a reference to a recent study about Arctic climate trend** (Jansen et al., 2020) **in the discussion section of the main manuscript.**

Please add a table with the specifications for the TanDEM-X database used in the study.

**R: We added Table S2 to the supplement which includes metadata of the used TanDEM-X acquisitions.**

**4) Response to reviewer 2**

This manuscript provides new geodetic estimates of glacier mass balance for the three main Russian Arctic archipelagos (Novaya Zemlya, Severnaya Zemlya, Franz Josef Land) and briefly discusses the results. The two most novel aspects of the study are that near-complete coverage of glacier elevation changes is obtained, and that the results indicate an increase in mass loss compared to earlier periods and studies.

The authors use digital elevation models (DEMs) derived from SAR interferometry of the TanDEM-X mission. This has the advantage of providing near-complete repeat coverage of glacier areas (93%), but can suffer from variable X-band radar signal penetration in snow/ice between satellite acquisitions. This is one of the main discussion points of the paper, and a correction-scheme is proposed for Novaya Zemlya where seasonal acquisition times were most different. Meteorological reanalysis data and supplementary DEM analyses are presented to support the approach, and results are provided both with and without penetration correction, as well as for two different density assumptions in the conversion between volume and mass change.

The main results appear plausible and relatively robust overall, but the differences related to acquisition times on Novaya Zemlya are puzzling and do not give a strong justification for the applied correction scheme. The potential magnitude and mechanisms of seasonal penetration differences are not well described or discussed, and the relevant parts of the manuscript (mainly Section 2.3) brings more confusion than clarity. For example, the paper does not say anything about the spatial coverage of the autumn and winter data of 2016/17 (Do they cover areas of potential different glacier change? Is there any overlap so that the two periods can be compared directly?) or if winter snow is partly accounted for in the co-registration process over land areas, which would limit the need for seasonal correction. See the specific comments below for further details on this issue.

The manuscript is written in a Brief Communication format, which is probably related with the authors' previous publications with similar methodology in other glacier regions, but I think that the present version suffers from too short/unclear methodology and very limited discussions. I think a lot of this can be fixed with improved writing and referencing, and perhaps by moving parts or all of Sections 2.2 (uncertainty assessment) and 2.3 (Dem Acquisition date correction) to the Supplement as these two sections are not satisfactory in the present form (see specific comments below). Alternatively, the manuscript could be expanded to a normal paper by making more complete data/methods sections and expanding the discussion of observed glacier changes which is now very brief. In any case, some major revisions are needed regarding these aspects.

**Initially we would like to thank the reviewer for the detailed and comprehensive comments.**

**Concerning the "Brief communications" format, we decided to use this short type of manuscript because the presented method and datasets have been described in a number of previous publications and the only significant changes are related to the temporal offsets between DEM acquisitions on Novaya Zemlya. However, we agree that the description of the workflow suffered from the short format. Therefore, we moved the description of the interferometric DEM creation and associated uncertainty section, as suggested, to the supplement because those chapters follow**

closely our previous publications. We also extended those method sections and inserted additional references which are related to the processing workflow. Within the main manuscript, we extended the description and discussion of radar signal penetration (new chapter 2.2 and 2.3) and included further figures (Fig. 1a/b, Fig. S2).

As suggested, we also revised the correction for temporal offsets of the DEM acquisitions on Novaya Zemlya:

- We included an analysis of differences between backscatter intensities during different acquisition months. The observed hypsometric distribution of backscatter intensity (revised manuscript: Fig. 1a & Fig. S1) indicates significant changes in surface conditions for the September 2016 acquisitions while backscatter intensity of the other acquisition months is relatively similar (Fig. 1).
- To estimate the vertical difference between September and winter acquisitions, we derived differences in surface elevations and respective backscatter from glacier areas which were acquired in September and winter 2016/17 (Fig. S2, ~3000 km²). The extracted vertical difference and respective September backscatter intensities are used to fit a linear regression model (chapter 2.2 & 2.3).
- Thereafter, the model is transferred to all glacier areas which were only acquired in September 2016 and the elevation change rate is adjusted accordingly. The overlapping (reference) glacier areas and those areas which were only acquired in September 2016 are indicated in Fig. 2a (revised manuscript: Fig. S2a). Fig. 2b shows the applied vertical correction values (Fig. S2b).

[Figure]

**Fig. 1 Backscatter intensity versus elevation of TanDEM-X acquisitions on Novaya Zemlya (2016/17). Black lines indicate the average backscatter intensity of each acquisition month.**

[Figure]

**Fig. 2 a) Overview of glacier areas which were acquired by TanDEM-X during September and winter 2016/17 (red dots) and areas which were measured only in September 2016 (blue triangles). b) Estimated vertical offset between DEM acquisitions of September and winter 2016/17.**

**The applied correction and further details of the extracted elevation and backscatter values are described in the revised chapter 2.2 and 2.3 and Fig. 1.**
**Please find our point-by-point responses below:**

Specific comments and edits:

Title: Since parts of Siberia is often considered to be in the Russian Arctic and there are areas with small mountain glaciers there, it would be more precise to say "Russian High Arctic" or "Russian Arctic archipelagos" in the title and elsewhere in the manuscript. Also, I think that "increased" is a more correct term than "accelerated" considering your results in relation to other studies.
**R: The term "Russian Arctic" in the manuscript refers to the regional subdivision of the Randolph Glacier Inventory which comprises the archipelagos of Franz Josef Land, Severnaya Zemlya and Novaya Zemlya as "Russian Arctic". But we agree that the term might be confusing and replaced it**

**with "Russian High Arctic" in the title and abstract. Additionally, the title was changed to: "Increased glacier mass loss in the Russian High Arctic (2010-2017)"**

L7: I assume you mean "atmospheric warming" or "surface warming", not the thermal state of the glaciers.
**R: Yes, included "atmospheric"**

L15: This reference only considers one region. Please provide a few other similar refs or a more general one covering multiple regions. Russian Arctic
**R: We included some other studies which focus on (increasing) glacier mass loss during recent years (~ >2010): (Zheng et al., 2018; Ciracì et al., 2020)**

L21: Or more broadly: "...and various corrections related to surrounding oceans, surface hydrology and glacial isostatic adjustment (GIA)."
**R: Sentence changed accordingly**

L30: What is the CoSSC tile product? Write out the acronym as a minimum.
**R: Included: "…Coregistered Single look Slant range Complex (CoSSC)…" which is the product specification by the data provider of TanDEM-X.**

L30: "Compared with..." – what do you actually mean? "Unlike..." or "Similar to..."
**R: In previous studies on glacierized regions outside the Arctic we used the SRTM DEM as reference surface while in the Arctic we applied the TanDEM-X Global DEM because SRTM was not acquired beyond 60°N. We therefore changed the beginning of the sentence to "Unlike previous studies (), …"**

L34: Did you cross-check this coastline against the glacier inventory to make sure no glacier areas were excluded? Please specify in the text to make this clear.
**R: Yes, the OpenStreetMap coastline was visually inspected and adjusted in areas where it did greatly differ from glacier areas of the Randolph Glacier inventory. Most changes were related to the glacier tongues of marine-terminating glaciers which also changed since the acquisition of the Randolph glacier inventory (see comment L40). Also, a small inverse buffer was applied to the coastline to account for an insufficient separation between land (stable ground for co-registration) and ocean/sea ice on some of the smaller islands of Franz Josef Land and Severnaya Zemlya. We added a respective explanation in the methods section of the supplement.**

L35: This relates to the sentence at L30. Please combine similar content at one place.
**R: Combined content with first sentence of chapter 2.1**

L36: Somewhat unclear. After a few reads I understand it as .... 2010/11 co-registered to Global DEM and mosaiced ... then 2016/17 co-registered to the 2010/11 mosaic to make a 2016/17 mosaic. Please clarify the text.
**R: Yes, extended & clarified the explanation. The DEM-creation methods section was moved to the supplement and extended.**

L37: I understand this as dividing by decimal numbers of years according to the dates of the source tiles. But that's confusing since you are differencing DEM mosaics. Does that mean you also made a mosaic layer of time differences? Or did you divide by an integer number of years (6) everywhere which would make more sense in a climatic mass balance perspective? Either approach could be justified, but this not discussed at all although it could have a significant impact on the results.
**R: Yes, a mosaic layer of time differences is created alongside the 2010/11 and 2016/17 DEM mosaics. This layer provides for each raster cell the exact time difference (as decimal number of years) between the acquisitions. We use this to calculate an individual elevation change rate (m/a)**

**for each elevation change value with the respective start and end date. We included this in the extended supplement methods.**

L39: Would be good to refer Fig. 1 here since the altitude dh/dt function is shown there.
**R: Included reference to Fig. 1**

L40: Isn't the inventory applied earlier than this, e.g. for the void filling? Also, the inventory is somewhat outdated, so what was done (or not) for glaciers that have undergone major changes such as the advancing Vavilov ice cap. The altitude-dependency of dh/dt in Fig S2 indicates that the Vavilov advance has been accounted for, whereas the less negative dh/dt of the lowermost altitudes of land-terminating glaciers in NZ indicate an impact from retreat which shouldn't influence overall mass rates (Gt/y), but could impact the area-specific rates (m/y). A brief discussion of these matters would be good to have somewhere in the manuscript. Note that there is a newer inventory for Novaya Zemlya (Rastner et al., 2017) which could be relevant for context or comparison.

**R: The Randolph Glacier Inventory of the Russian Arctic archipelagos was created from optical images between 2000 and 2010 but there is no specific timestamp provided within this period for a number of glaciers. We made some manual adjustments as the retreat of some of the major (marine-terminating) outlet glaciers and of course the surge of the Vavilov ice cap were not covered by the original inventory.**
**A comparison with the recent inventory for Novaya Zemlya (Rastner et al., 2017) also indicated that most changes in glacier outlines are related to the retreat of outlet glaciers along the coastlines. The total glacier areas of Novaya Zemlya provided by the Randolph inventory (~22,128 km²) and Rastner et al. 2017 (~22,379±246 km²) are very similar.**
**Unfortunately, there are no other recent inventories which cover the remaining glacier areas of Severnaya Zemlya and Franz Josef Land. Therefore, we decided to use the (modified) Randolph inventory as it provides a homogeneous glacier area dataset for the entire region.**
**The less negative elevation change rates of the lowermost elevation bins are related to glacier retreat during the observation period and the temporal offset between outlines and DEM (we also included this in the caption of Fig. S3). It is not possible to update the entire inventory due to a lack of cloud-free images in this region.**
**We included a small section in the supplement methods to describe the applied glacier inventory.**

L42-43: It's not the scenarios that change, but the firn pack. Rewrite sentence to make it clear what you actually mean here. Also, do you consider this issue to be within the error estimates you provide or as something that comes on top of that (i.e. not considered).
**R: Changed sentence to "Possible changes in the glacier ice density (e.g. firn compaction) …" (supplement methods, Line 64-65).**
**The suggested uncertainty of ±60 kg m-3** (Huss, 2013)**, which is included in our uncertainty estimate, is recommended for observation periods of more than 5 years, the presence of firn and volume changes different from zero. However, the mentioned study reported that this mean conversion factor can significantly vary under different conditions. As there are no observations of glacier density in the Russian High Arctic, we cannot quantify a region-specific uncertainty value for the volume to mass conversion.**

L44: Unclear and not strictly correct. It does include frontal melt/calving when that balances the ice outflux, but it does not include subaqueous glacier volume changes related to advance or retreat. This should be made clear, and also its potential relevance for the overall glacier mass balance and sea-level contribution, here or in the discussion.
**R: Rewrote sentence (supplement methods, Line 66-67).**

L45: The uncertainty section is not understandable by itself and needs to be rewritten. There are parameters that are not fully explained, units are unclear, and it is hard to follow the logic unless a lot of time is spent with Table S1 and given references.
**R: We moved the uncertainty section to the supplementary materials and extended the description of the applied workflow and equations.**

Eq. 1: Is this equation from previous work or is it unique for this study? It appears like mass rate uncertainty is a factor of the mass rate itself which does not make sense to me if the mass rate turn out to be near zero.
**R: Equation 1 is from previous studies, e.g. (Braun et al., 2019; Seehaus et al., 2019) and was only slightly modified for this study because we added an estimate of the signal surface penetration (-> winter to autumn acquisitions) directly to the elevation change uncertainty (and thereby also to the mass change uncertainty). In previous studies, surface penetration was estimated as a "bias volume" and thus only included in the volume/mass change uncertainty.**

L56: Is Sg ever larger than Scor here? If not, then it's confusing to include this equation. I understand it as you are calculating errors per region, not per glacier.
**R: Yes, this part is for the large ice bodies of the Russian High Arctic not relevant. Still, we would like to keep the entire equation in the methods section because of consistency with previous publications of the presented uncertainty calculation.**

L60: How was this number found? Not clear from Section 2.3. It is also unclear if the approximate 2 m penetration difference (Spen) is applied only to the NZ autumn data or to all data in all regions which would make most sense.
**R: This number was the originally determined offset value between autumn and winter acquisitions on Novaya Zemlya. We changed the respective analysis and descriptions in the text (and in the supplement methods). The revised vertical offset value for Novaya Zemlya (~ 2 m) is derived by an analysis of backscatter intensity on overlapping glacier areas (chapter 2.2 & 2.3 in revised manuscript). For the elevation change uncertainty, we applied this value to Franz Josef Land and Novaya Zemlya but weighted it with the respective autumn area because the difference in surface penetration is expected to be small or zero for acquisitions from the same season. For Severnaya Zemlya we used an estimate of average penetration depth because all DEMs were acquired in the same season (see supplement methods section).**

L64-79: I like the comparative elevation differencing from winter 2010/11 to autumn (WA) and winter (WW) 2016/17, respectively, and I agree it might be the best way to try to account for errors related to signal penetration, but the logic is too simplified. Is it just melting or non-melting surface condition that is relevant? Widespread melting conditions are unlikely after mid-September, and ERA5 is too coarse to capture topographic temperature variations. In that context, I would consider differences in SAR backscatter to be relevant. And how deep can the X-band signal penetrate? There is no mention or references regarding that. For example, is the last summer-surface a dominant reflection horizon during winter or can it penetrate even deeper. In the latter case, the meteorological conditions of previous years might also matter.
**R: The depth of surface penetration of the X-band radar strongly depends on the prevailing surface conditions during the DEM acquisition. In general, penetration is low for melting conditions and high for dry and frozen surfaces. For the DEM-differencing in this study, the relative difference between the penetration depths of the acquisitions at the beginning and end of the observation period is relevant. It is likely that this difference is small or zero for acquisitions of similar seasons or dates but increases when comparing DEMs of different seasons. Therefore, we included a more specific analysis of local radar backscatter intensity and the related differences in measured surface elevation (revised chapter 2.2 & 2.3) to account for the temporal offsets between acquisitions of**

**autumn and winter 2016/17 (see response to general comments). Additionally, chapter 2.2 includes now a general description of signal penetration and respective references.**

L84: Fig. S2 shows altitude dependency, not whether a glacier is small or large. Rephrase or refer to Fig. 1 instead where it does seem like the largest glacier fronts thin the most.
**R: Changed figure reference to Fig. 2 (former Fig. 1).**

L94: Unclear. Rather something like this: "Relations between acquisition times, monthly temperatures and derived elevation change rates for NZ are shown..."
**R: Rephrased/changed this part of the results section.**

L98: Redundant wording; elevation gains are always positive.
**R: Removed "positive"**

L102-104: True if no penetration, whereas if fresh cold snow is transparent then it can be considered as autumn 2010 to autumn 2016 changes, with no seasonal snow bias.
**R: The part about signal penetration in the discussion section was rewritten and extended. The discussion of potential offsets in measured winter accumulation or signal penetration differences has been extended. We also included the radar backscatter as indicator of changing surface conditions between September 2016 and winter 2016/17.**

L112: This is also what I speculated (see previous comment), but then dh/dt from the WA and WW periods should have been more or less similar, which is not the case.
**R: please see comment above**

L113-115: I don't understand the logic here. Are you suggesting penetration into the firn/ice during winters and near-surface reflection during autumn? If so, you are in practice measuring a "delayed mass balance" (shifted backwards in time).
**R: Yes, it is likely that the penetration in winter 2010/11 and 2016/17 was higher (but similar in both cases) while in September 2016 the measured elevations were closer to the actual glacier surface (less penetration). We extended and rewrote this part of the discussion.**

L117: The figure indicates largest warming for the northern islands (FJL and SZ) and smallest for the southern ones (NZ), which is opposite of what you say. But warming might still have a larger impact in the south since climate is in general warmer and closer to the melting point. The most relevant aspect for this paper would be how 2010-17 stands out from the longer-term climate, especially during the summer melt season. Any relevant references that have studied climate change in this region in more detail?
**R: To our knowledge there are no recent studies which analysed the Russian High Arctic specifically. We inserted a recent reference of climate trend analysis in the entire Arctic** (Jansen et al., 2020)**. In the revised manuscript, Line 117 was removed and combined with Line 159.**

L119: What about the comparable Wouters et al. (2019) paper?
**R: Added Wouters et al. 2019**

L121: How much of your mass loss is related to the surge of Vavilov ice cap? Would there be a substantial remaining mass loss if dynamic areas of Vavilov and Academy of Sciences ice caps were excluded? I miss such aspects of the discussion.
**R: The Glacier elevation change of the Severnaya Zemlya archipelago would be approximately half as negative without the outlet glaciers of the Vavilov and Academy of Sciences ice caps. We included this in Line 144-146.**

L123-l30: The study of Melkonian et al. (2016) is also very relevant for this discussion, considering both long-term elevation changes and ice dynamics.
**R: Included Melkonian et al. (2016), Line 150: "(Carr et al., 2014). Long-term observations also indicate a more rapid thinning during recent years, particularly at the termini of marine-terminating glaciers (Melkonian et al., 2016)"**

L129: are not always related to -> does not seem to be related to
**R: Ok, changed**

L132: Zhang et al. (2018) is also very relevant here (only referenced in the Supplement)
**R: Included (Zheng et al., 2018)**

L137: showed -> has shown
**R: Ok, changed**

L138: ...between 2010 and 2017
**R: Included**

L139: Unclear. Do you mean that Arctic glacier mass losses are increasing more than non-polar ones? If so, in total or specific rates?
**R: This sentence refers to the sea level rise contribution of different glacierized regions during the last decades. At the end of the 20th and beginning of 21st century, many Arctic glaciers showed small elevation changes or even balanced conditions. Their contribution to sea level rise was therefore rather small compared to glacier outside the Arctic which showed much higher melt rates. Various studies indicate that this pattern is changing in recent years and increasing melt rates are also measured in the polar region. While specific change rates of Arctic glaciers are still less negative than those of mountain glaciers outside the polar regions, the total mass loss (and therefore also the contribution to sea level) is higher due to the very large glacier areas.**

L140: You are basically listing all regions except Svalbard. Is this sentence needed?
**R: Removed sentence**

Fig. 1: Nice figure. Is it possible to also show the autumn (A) versus winter (W) coverage of DEMs in the 2016/17 seasons? Or in the supplement to keep this figure clean.
**R: We included another map of Novaya Zemlya in the supplement which shows glacier areas covered in autumn and winter 2016/17 (Fig. S2).**

Table S1: You seem to use AW here as an abbreviation for area-weighted, which is confusing because you use AW as an abbreviation for autumn-winter elsewhere in the manuscript. And at L51 you write slope-weighted instead of area-weighted.
**R: We removed the abbreviation for autumn-winter in the manuscript because the correction was changed to September areas only.**

Fig. S1: Are the climatological data extracted for the entire regions or specifically for the glacier areas? I don't think that is mentioned anywhere in the manuscript.
**R: The climate data used for the glacier regions was changed to the ERA5 Land product (which provides a better spatial resolution) in the revised version. We also added specifications about the extracted area in the caption and directly in the plots. The regional data was extracted with a bounding box with the extent of the glacier inventory of each archipelago. Ocean areas are not included.**

Fig. S4: Nice compilation of results. For FJZ, it should be Zheng et al. (2018), not 2019 which is another paper.
**R: Thank you very much, changed Zheng et al. (2019) to (2018).**

References

Melkonian, A. K., M. J. Willis, M. E. Pritchard, and A. J. Stewart (2016), Recent changes in glacier velocities and thinning at Novaya Zemlya, Remote Sens. Environ., 174, 244-257.

Rastner, P., T. Strozzi, and F. Paul (2017), Fusion of Multi-Source Satellite Data and DEMs to Create a New Glacier Inventory for Novaya Zemlya, Remote Sensing, 9(11).

**5) References**

Abdullahi, S., Wessel, B., Huber, M., Wendleder, A., Roth, A., and Kuenzer, C.: Estimating Penetration-Related X-Band InSAR Elevation Bias: A Study over the Greenland Ice Sheet, 19, 2019.

Braun, M. H., Malz, P., Sommer, C., Farías-Barahona, D., Sauter, T., Casassa, G., Soruco, A., Skvarca, P., and Seehaus, T. C.: Constraining glacier elevation and mass changes in South America, Nat. Clim. Change, 9, 130–136, https://doi.org/10.1038/s41558-018-0375-7, 2019.

Carr, J. R., Stokes, C., and Vieli, A.: Recent retreat of major outlet glaciers on Novaya Zemlya, Russian Arctic, influenced by fjord geometry and sea-ice conditions, J. Glaciol., 60, 155–170, https://doi.org/10.3189/2014JoG13J122, 2014.

Ciracì, E., Velicogna, I., and Sutterley, T.: Mass Balance of Novaya Zemlya Archipelago, Russian High Arctic, Using Time-Variable Gravity from GRACE and Altimetry Data from ICESat and CryoSat-2, Remote Sens., 10, 1817, https://doi.org/10.3390/rs10111817, 2018.

Ciracì, E., Velicogna, I., and Swenson, S.: Continuity of the Mass Loss of the World's Glaciers and Ice Caps From the GRACE and GRACE Follow-On Missions, Geophys. Res. Lett., 47, https://doi.org/10.1029/2019GL086926, 2020.

Huss, M.: Density assumptions for converting geodetic glacier volume change to mass change, The Cryosphere, 7, 877–887, https://doi.org/10.5194/tc-7-877-2013, 2013.

Jansen, E., Christensen, J. H., Dokken, T., Nisancioglu, K. H., Vinther, B. M., Capron, E., Guo, C., Jensen, M. F., Langen, P. L., Pedersen, R. A., Yang, S., Bentsen, M., Kjær, H. A., Sadatzki, H., Sessford, E., and Stendel, M.: Past perspectives on the present era of abrupt Arctic climate change, Nat. Clim. Change, 10, 714–721, https://doi.org/10.1038/s41558-020-0860-7, 2020.

Melkonian, A. K., Willis, M. J., Pritchard, M. E., and Stewart, A. J.: Recent changes in glacier velocities and thinning at Novaya Zemlya, Remote Sens. Environ., 174, 244–257, https://doi.org/10.1016/j.rse.2015.11.001, 2016.

Moholdt, G., Wouters, B., and Gardner, A. S.: Recent mass changes of glaciers in the Russian High Arctic: GLACIER MASS CHANGES, RUSSIAN ARCTIC, Geophys. Res. Lett., 39, n/a-n/a, https://doi.org/10.1029/2012GL051466, 2012.

Sánchez-Gámez, P., Navarro, F. J., Benham, T. J., Glazovsky, A. F., Bassford, R. P., and Dowdeswell, J. A.: Intra- and inter-annual variability in dynamic discharge from the Academy of Sciences Ice Cap, Severnaya Zemlya, Russian Arctic, and its role in modulating mass balance, J. Glaciol., 65, 780–797, https://doi.org/10.1017/jog.2019.58, 2019.

Seehaus, T., Malz, P., Sommer, C., Lippl, S., Cochachin, A., and Braun, M.: Changes of the tropical glaciers throughout Peru between 2000 and 2016 – mass balance and area fluctuations, The Cryosphere, 13, 2537–2556, https://doi.org/10.5194/tc-13-2537-2019, 2019.

Zheng, W., Pritchard, M. E., Willis, M. J., Tepes, P., Gourmelen, N., Benham, T. J., and Dowdeswell, J. A.: Accelerating glacier mass loss on Franz Josef Land, Russian Arctic, Remote Sens. Environ., 211, 357–375, https://doi.org/10.1016/j.rse.2018.04.004, 2018.

---

## Referee Report (RR1)

**2nd report for the Brief communication paper: **Increased glacier mass loss in the Russian Arctic (2010-2017)**

The Cryosphere Discuss. https://tc.copernicus.org/preprints/tc-2020-358/

This 2nd report refers to the revised manuscript **tc-2020-358-manuscript-version3.pdf** and **tc-2020-358-supplement-version3.pdf** from 26.05.2021

**Comments to the authors**

Thank you for responding to my comments and the changes implemented for improving the work. The paper is better structured now the Data and Methods section reveals relevant aspects specific to the used dataset on this particular glaciated region. Although, as suggested in the review of the first version, the analysis of the backscattering coefficients was added, the estimation of the penetration depth of the X-band SAR signal into the glacier volume is based on wrong assumptions.

I have some doubts regarding the correctness of Eq.1 by the following reasons (see also Dall, 2007): (i) The penetration depth (dp) refers to the vertical. (ii) For small relative penetration the elevation bias  $h_b$  can be approximated by the two-way power penetration depth:  $dp_2 = dp/2 \approx h_b$ . (not by the one-way penetration depth). (iii) For given InSAR geometry and propagation conditions (permittivity) dp is related to the oblique radar propagation path multiplied by the cos of the refraction angle in the snow volume.

The penetration bias depends not only on the radar wave propagation properties in the snow volume but also on the interferometric baseline and incidence angle. The impact of these parameters needs to be considered if an observed elevation bias (or penetration value) is applied to another InSAR scene.

Regarding the Fig. 1 and the related text (line 70 and below):

For estimating the penetration-related elevation bias in Eq. 2 the difference in  $\sigma^0$  between September (surface melt) and mean  $\sigma^0$  of Oct. to Jan. is used as proxy. This implies an immediate switch for melting state in Sept. to dry snow with deep penetration in Oct. In reality this transition is gradual in time which means using October (and possibly also November) data in the "winter" ensemble causes a bias for estimating the penetration for the winter case. In Fig. 1a the Oct. and Nov.  $\sigma^0$  values are lower than the Jan. values (in particular in the 300 m to 600 m elevation zone).

Fig. 1a: The used procedure (calibration coefficients) to convert amplitude to  $\sigma^0$  needs to be checked because as far as I see  $\sigma^0$  values are down to -30 dB which is far below NESZ. I also miss mentioning in the paper or supplement in which way was the incidence angle dependence of backscatter intensity taken into account. Also, the look angle of the various TanDEM-X acquisitions is not given anywhere (Table S2 gives a list but with some redundant information). In particular for wet snow  $\sigma^0$  show large changes with the incidence angle. Results (line 92 and below and Fig. 2)

The error bar is decreasing with the decreasing magnitude of  $\Delta h/\Delta t$  and increasing elevation. Usually, the geodetic error should be independent on  $\Delta h/\Delta t$ . At higher elevations where  $\Delta h/\Delta t$  small additional error contributions may be added resulting in larger error bars than at the termini. I recommend therefore to revise the error calculations. Regarding the uncertainty assessment for the mass change (now equation (1) in Supplement in the current version of the manuscript) I also have some doubts (expressed also by reviewer #2). According to this equation the error of the mass change estimate depends on the mass change magnitude  $\Delta M/\Delta t$ . This would mean a zero mass change estimate would yield a perfect result (no error). But then the first term of the sum would compensate: small (near zero)  $\Delta h/\Delta t$  leads to very large error and vice versa (in case of large mass changes). These terms contributing to the error budget should be treated independently to hold for quadrature sum. See also (Nuth & Kääb, 2011).

**Specific comments**

Main paper:

Line 39 into the glacier volume.

Line 41 increases in dry snow.

Line 51 snow and ice properties at the glacier surface can have significant impact on ...

Line 58 much lower backscatter values then ...

Line 64 and 66 replace "surface penetration" by penetration into the volume

Line 86 smaller than on NZ

Line 89 Fig S1a

Line 298 Fig. 1a (identical with Fig S1e): Mean backscatter corresponding to 2016-12 is not visible

**Supplement**

Adding Table S2 is welcome but contains a lot of redundant information and not the important one. One row pro TanDEM-X acquisition (instead of one row pro CoSSC framing of the same datatake) would be enough but some additional information would be useful: Beff, HoA, incidence angle, etc similar to other publications using self-processed TanDEM-X DEMs (e.g. Table 1 in (Malz et al., 2018)). Keep the established acronyms and labels used in the metadata: Active sensor instead of "transmitting", "Strip" should be "Beam" and TSX-1 and TDX-1 (instead of TST and TDT), Relative orbit instead of "Path number".

Line 122 quadrature sum

**References:**

Dall, J.: InSAR elevation bias caused by penetration into uniform volumes, IEEE Trans. Geosci. Remote Sens., 45, 2319–2324, 2007

Malz, P.; Meier, W.; Casassa, G.; Jaña, R.; Skvarca, P.; Braun, M.H. Elevation and Mass Changes of the Southern Patagonia Icefield Derived from TanDEM-X and SRTM Data. *Remote Sens.* **2018**, *10*, 188. https://doi.org/10.3390/rs10020188

Nuth, C. and Käáb, A.: Co-registration and bias corrections of satellite elevation data sets for quantifying glacier thickness change, The Cryosphere, 5, 271–290, https://doi.org/10.5194/tc-5-271-2011, 2011.

---

## Referee Report (RR2)

**3rd report for the Brief communication paper: Increased glacier mass loss in the Russian Arctic (2010-2017)**

The Cryosphere Discuss. https://tc.copernicus.org/preprints/tc-2020-358/

This 3rd report refers to the revised manuscript **tc-2020-358-manuscript-version4.pdf** and **tc-2020-358-supplement-version4.pdf** from 21.09.2021

Green: Response of the authors to reviewer 2
Blue: Report #3

**Comments to the authors**

Thank you for responding one more time to my comments and the changes implemented for improving the work. I address here the critical issues remaining to be clarified.

*As suggested, we applied the approach using the two-way power penetration to estimate the surface penetration depth instead of the trigonometric function. To estimate the refraction angle into the glacier surface, we referred to a reference study on in-situ experiments in Antarctica (see below). Using this approach, the following paragraphs would replace the former Eq. 1 (L.70) in the revised manuscript):

"The vertical differences between heights of autumn and winter DEM acquisitions are converted into depths of signal penetration into the glacier volume using Eq. 1 following (Dall, 2007):

$$l = \frac{d_p}{\cos(\Theta_v)} \; ; \; d_p = 2 \times h_b \qquad \text{Eq. 1}$$

where $l$ is the penetration length and $\Theta_v$ the refraction angle into the volume. $d_p$ is the two-way power penetration depth and can be approximated by two times the vertical elevation bias $h_b$ (Dall, 2007). To derive the refraction angle ($\Theta_v$), Eq. 2 (Snell´s law) is applied:

$$\sin(\Theta_v) = n_1 \times \frac{\sin(\Theta_v)}{n_2} \qquad \text{Eq. 2}$$

where $\Theta_l$ is the local incidence angle, $n_1$ the refractive index of air (1.000293) and $n_2$ the refractive index of glacier ice. For the permittivity of ice, various values have been reported in literature (Rasmussen, 1986; Dowdeswell and Evans, 2004). In general, the refractive index of ice increases with depths due to changes in density. Therefore, we refer to a detailed in-situ study on refraction measurements from the ice surface down to depths of 150m in Antarctica (Kravchenko et al., 2004). For glacier ice close to the surface (0 to -40 m depth), they found values between ~1.3 and ~1.5 as index of refraction. Thus, we apply a refractive index of ice ($n_2$) of 1.4 as the approximate permittivity of ice close to the glacier surface."

I would have preferred a revision in the manuscript directly to avoid confusion. The change of Eq.1 after (Dall, 2007) is welcome. In this paper the vertical penetration bias ($h_b$) is approximately the two-way power penetration depth (dp2) in the case of a small penetration compared to the height of ambiguity (eq. 13 in (Dall, 2007)). With dp the **one-way** power penetration depth is denoted.

But since you already have an observed height difference I suggest to derive the correction directly. In the manuscript dp is once defined as depth of penetration into the volume (line 61) and below (line 68) as penetration bias. These are not the same. In this case I guess EQ2 refers to the penetration related elevation bias $h_b$.

In Author's response Eq.2 right: sin $\Theta_l$

But you don't need to apply Snell's law if the penetration is defined as vertical. If yes, please explain. Besides Kravchenko et al. report on measurements of the dielectric permittivity on the South Pole, with a density profile of cold polar firn, very different from that of the percolation zone of Arctic glaciers. The real part of permittivity of dry snow and firn can be computed from the density with a slightly non-linear relation (Ulaby and Long, 2014). Typical density profiles from the percolation zone of Artic glaciers have been reported in several papers (e.g. for Svalbard by Marchenko et al., 2017).

We also recalculated the signal penetration corrected elevation & mass change for Novaya Zemlya with this approach but the results are almost exactly the same as in the original version (original $\Delta h/\Delta t$ = -0.643 m/a and new $\Delta h/\Delta t$ = -0.644 m/a). The only significant change would be the estimated average vertical offset (3.5 m instead of 2.13 m).

However, we are not sure if this approach improves the accuracy of the estimate. The model presented by (Dall, 2007) assumes an infinite volume, i.e. the microwave signal is not scattered by any layer below the surface (volume scattering). While this might be the case for some glacier areas on Novaya Zemlya with rather high vertical offsets, it is not unlikely that there is a scattering layer below the actual surface (e.g. melt/refrezzing of a previous summer) for glacier areas with smaller vertical differences. For those areas, the new approach could overestimate the penetration depth and produce a rather high average penetration depth for all September acquisitions.

For those reasons, we did not include the two-way power penetration estimate into the revisedmanuscript yet, but we would be very interested to hear the reviewer´s opinion regarding those concerns.

EQ2 (line 70) should refer to the $\Delta h_{W-A}$.(or $h_b$) A plot of its altitude dependence over the overlapping areas (red dots in Figure S2a) would be crucial to understand how this regression was derived. The regressions shown in Fig 1c and Fig 1d are not explained at all. Hard to understand how they contribute to EQ2.

 *"We did not adjust for differences in incidence angle or effective baseline because the viewing geometries of the majority of the used SAR acquisitions are rather similar (Table S2). For 99% of the glacierized area ofNovaya Zemlya, the difference in incidence angles is not larger than 2° (39.3° - 41.3°) while for 93% ofarea the average baseline is 91.9 m (87.8 m – 95.4 m)."*

Accepted.

*"It is noteworthy, that the applied regional correction scheme can introduce a larger uncertainty at a local glacier scale caused by different surface and backscatter conditions between the specific TanDEM-X acquisitions (Fig. S2b). However, due to the limited extent of overlapping glacier areas (Fig. S2a), it is not possible to derive a date-specific intensity correction foreach DEM strip. Thus, the applied linear model does rather represent an average difference in surfacepenetration depth between autumn and winter SAR data."*

Accepted.

*The intensity images are created using the Gamma remote sensing software environment (Werner et al., 2000). The radiometric calibration of the amplitude to σ0 values is automatically performed by the conversion algorithm from the CoSSC to the Gamma data format (using the metadata of the CoSSC data product). The respective algorithms are part of the interferometry (ISP) module and described in the *Interferometric SAR Processor – ISP user´s guide* (GAMMA Interferometric SAR Processor (ISP), 2021) in section 2.2.7 (TerraSAR-X & TanDEM-X data read algorithms) and 2.4.5 (radiometric calibration procedure). A link to this user guide is provided in the reference list.

Still the backscattering values used for the signal penetration (Fig 1c and Fig 1d) are going down to -27 dB. The noise level (NESZ) for the beams around 37-41 deg incidence is around -24 dB. It is annotated in each CoSSC product. This questions additionally EQ2.

*Concerning the incidence angle, we revised Table S2 following the suggestions (see respective comment below) and extended the methods section (see second comment).
Accepted.

*The hypsometric bars shown in Fig. 2 refer to the normalized median absolute deviation of $\Delta h/\Delta t$ measurements on glacier areas within each elevation bin. Therefore, the bars are largest a low elevations because the spread of measured $\Delta h/\Delta t$ values is large due to the presence of strong thinning glacier termini. At high altitudes, the range of measured $\Delta h/\Delta t$ values is in general much smaller (see also $\Delta h/\Delta t$ maps of Fig.2) and thereby also the bar. We extended the caption of Fig. 2 because the description of the shown bars was missing. Regarding the geodetic error, we did not calculate a mass change error for each elevation bin but for the entire region (based on the mean regional elevation change and respective uncertainty).

The errors in dh/dt [m/yr] related to vertical co-registration do not depend on the magnitude of retrieved dh/dt, but on the uncertainty in co-registration, independent of the magnitude. Even zero dh/dt has the same error in respect to vertical co-registration. For the higher elevation zones (firn areas) where penetration-related errors are added, the error in the elevation change rate dh/dt [m/yr] should by higher in the firn areas than in ice areas of glaciers. Fig. 2 d to f: Please explain the term "normalized median absolute deviation of elevation change measurements of each elevation bin". An equation would be helpful.

*Regarding to Eq. 1 (Supplement), unfortunately we do not quite understand the question referring to the mass change and elevation change uncertainty: The first term of the sum is the ratio between the $\Delta h/\Delta t$ uncertainty ($\delta_{\Delta h/\Delta t}$) and the mean (glacier) $\Delta h/\Delta t$ estimate. While the $\Delta h/\Delta t$ estimate is derived on glacierized areas, the $\Delta h/\Delta t$ uncertainty ($\delta_{\Delta h/\Delta t}$) mainly indicates the potentially remaining offsets on non-glacier areas after the co-registration (and also other sources of uncertainty, Supplement Eq.2). If $\Delta h/\Delta t$

would be very small (and thereby also ΔM/Δt), the uncertainty of ΔM/Δt could still be relatively high if $\delta_{\Delta h/\Delta t}$ (off-ice) is high compared to Δh/Δt (on-ice). For example, the measured elevation change rate is rather small but there are a lot of artificial elevation offsets remaining after the co-registration. In this case $\delta_{\Delta M/\Delta t}$ would be high compared to ΔM/Δt.

Accepted. For Eq. 1 (Supplement) there was some misunderstanding because of the same symbol (delta) used for relative error (right hand side) and absolute error (left hand side). The error estimate refers to the total change of mass over an extended area (though not defined; should be explained). There is one (rather unlikely case), when the equation is wrong: if the retrieved Δh/Δt is exactly zero. According to Eq. 1 this yields zero error for the mass change.

*Fig. 1a shows a subset of backscatter values (5000 random samples) because otherwise the figure would be too busy. The acquisitions of December 2016 cover only a very small fraction of the Novaya Zemlya ice cap (~10 km²). For this reason, there are only very few December datapoints visible and the mean backscatter was only calculated and plotted for the lowest elevation bin. The mean value is almost the same as for October 2016 (triangle) and therefore difficult to identify in the figure.
Now I can see it. Thanks for the explanation!

*We changed Table S2 and included only one row per TanDEM-X acquisition instead of each CoSSC frame. The terminology was adjusted and the new columns include now, as suggested, acquisition date, acquisition start time, active satellite, orbit direction, relative orbit, strip length (number of CoSSC frames), effective baseline, height of ambiguity and incidence angle.
Accepted.

**References:**
Dall, J.: InSAR elevation bias caused by penetration into uniform volumes, IEEE Trans. Geosci. Remote Sens., 45, 2319–2324, 2007

Ulaby, Fawwaz & Long, David & Blackwell, William & Elachi, Charles & Fung, Adrian & Ruf, Christopher & Sarabandi, K. & Zyl, Jakob & Zebker, Howard. (2014). Microwave Radar and Radiometric Remote Sensing.

MARCHENKO, S., POHJOLA, V., PETTERSSON, R., VAN PELT, W., VEGA, C., MACHGUTH, H., . . . ISAKSSON, E. (2017). A plot-scale study of firn stratigraphy at Lomonosovfonna, Svalbard, using ice cores, borehole video and GPR surveys in 2012–14. Journal of Glaciology, 63(237), 67-78. doi:10.1017/jog.2016.118

---

## Author Response (AR2)

**Author´s response to 2ⁿᵈ reviews of:**

**Brief communication: Increased glacier mass loss in the Russian High Arctic (2010-2017)**

Christian Sommer[1], Thorsten Seehaus[1], Andrey Glazovsky[2], Matthias H. Braun[1]

[1]Institut für Geographie, Friedrich-Alexander-Universität Erlangen-Nürnberg, Erlangen, 91058, Germany
[2]Institute of Geography RAS, Moscow, 119017, Russia

*Correspondence to*: Christian Sommer (chris.sommer@fau.de)

**Table of Contents**

**Response to the editor:**

*Editor report*

Comments to the Author:

Dear authors,

I now collected two reviews of your revised manuscript.

In general, both reviewers are rather positive about your revisions. They acknowledge your efforts and agree that the manuscript has been improved compared to the initial submission. However, they still suggest some significant improvements are needed before publication. In particular, reviewer#2 challenges some of the corrections and ask to double check some of the hypothesis/equations that you used.

*Thank you very much for the extension which was very helpful to discuss some of the suggested revisions.

Following the suggestions of reviewer 1, we made some structural changes to the methods chapter and extended some parts of the discussion (please see detailed comments).

We also applied the different conversion approach between vertical height differences and surface penetration depths which was suggested by reviewer 2. The new calculation does not change the overall elevation change rate but affects the estimated depth of signal penetration. However, we are not entirely convinced if this approach is an improvement to our previous version because it is based on some assumptions which might be only correct for some areas on Novaya Zemlya. Particularly, it assumes radar penetration into an infinite volume (i.e. no reflection from a previously melted and refrozen surface layer). Therefore, we described the new calculations and results in our reply to reviewer 2 but did not change the respective paragraphs in the revised manuscript yet. We would be interested to hear the reviewer´s opinion on this alternative calculation and our concerns regarding the involved assumptions.

In addition, you could compare your estimate to the recently published Tepes et al., (10.1016/j.rse.2021.112481) using CS-2 for the 2010-2017 time period. A figure comparing the available estimates for the Russian Arctic (as done in Tepes et al.) would be welcome to put your findings into context (in the supplement if you lack space for it). It seems that your mass balance estimates is more negative than all previous estimates for the Russian Arctic, something you may want to comment on.

*Yes, the available mass change results for the Russian Arctic vary between different studies and even measurements of the same sensors (e.g. different GRACE estimates for the Russian Arctic in Fig. S5). The TanDEM-X results (> 2010) are similar to the measurements of e.g. (Ciracì et al., 2020; Zheng et al., 2018; Ciracì et al., 2018) but rather high compared to (Hugonnet et al., 2021; Tepes et al., 2021). Yet, an increase in mass loss during the 21[st] century is shown by most studies. We also mentioned those differences and similarities in the discussion section but cannot provide a definite explanation for the different results. Regarding the recent CryoSat-2 study, we included Tepes et al. (2021) in section 4 (Discussion of mass change results) of the revised manuscript. Also, there is already figure S5 in the supplement (now also including Tepes et al.) which shows available results from the Russian Arctic and individual archipelagos. Or did you mean a different kind of figure? If so, we could still include it in the supplement.

Author´s response: Increased glacier mass loss in the Russian High Arctic (2010-2017)

Note that I am away until 8 August so no rush to submit your revised text (even if the system is pressing you).

We are now looking forward for a revised manuscript together with a point-by-point response.

Best regards,

Etienne Berthier

*References*

Ciracì, E., Velicogna, I., and Sutterley, T.: Mass Balance of Novaya Zemlya Archipelago, Russian High Arctic, Using Time-Variable Gravity from GRACE and Altimetry Data from ICESat and CryoSat-2, Remote Sens., 10, 1817, https://doi.org/10.3390/rs10111817, 2018.

Ciracì, E., Velicogna, I., and Swenson, S.: Continuity of the Mass Loss of the World's Glaciers and Ice Caps From the GRACE and GRACE Follow-On Missions, Geophys. Res. Lett., 47, https://doi.org/10.1029/2019GL086926, 2020.

Hugonnet, R., McNabb, R., Berthier, E., Menounos, B., Nuth, C., Girod, L., Farinotti, D., Huss, M., Dussaillant, I., Brun, F., and Kääb, A.: Accelerated global glacier mass loss in the early twenty-first century, Nature, 592, 726–731, https://doi.org/10.1038/s41586-021-03436-z, 2021.

Tepes, P., Gourmelen, N., Nienow, P., Tsamados, M., Shepherd, A., and Weissgerber, F.: Changes in elevation and mass of Arctic glaciers and ice caps, 2010–2017, Remote Sens. Environ., 261, 112481, https://doi.org/10.1016/j.rse.2021.112481, 2021.

Zheng, W., Pritchard, M. E., Willis, M. J., Tepes, P., Gourmelen, N., Benham, T. J., and Dowdeswell, J. A.: Accelerating glacier mass loss on Franz Josef Land, Russian Arctic, Remote Sens. Environ., 211, 357–375, https://doi.org/10.1016/j.rse.2018.04.004, 2018.

**Response to reviewer 1:**

**Suggestions for revision or reasons for rejection (will be published if the paper is accepted for final publication)**

This manuscript has been heavily revised with a new SAR-signal penetration correction based on an empirical relation between seasonal SAR backscatter and measured elevation differences. The correction scheme lacks proper validation (e.g. by satellite altimetry), but the authors provide both corrected and uncorrected results which are not so different that they alter the main findings and conclusions. So all in all, I think they have succeed to address the main issues at an appropriate level, providing robust results of recent glacier changes across the Russian High Arctic at a detailed level not presented before.

The manuscript has also been restructured by moving most of the standard methodology (established in earlier publications) to the supplement instead of the confusing and partly repetitive split-up of the previous manuscript version. This has allowed more details on the penetration issue to be included in the main manuscript. Both aspects are clear improvements, but as indicated in the chronological comments below, I think the manuscript can still benefit from some smaller restructuring and inclusion of essential methodological details such as density conversion factors and the parallel calculation of mass-change rates with and without penetration correction. There are also some unclear sentences and inconsistent terminology, so I recommend the authors to do a careful language read-through and edit for the final manuscript version.

*Thank you very much for the 2$^{nd}$ review. We added some description to the methods section including density conversion and the calculation of mass change rates with and without signal penetration correction. Additionally, we made some changes to the structure of the methods and results chapter. As suggested, we merged the two sections about surface penetration in the methods chapter. Please find our detailed responses below (Line numbers refer to the track-changes version):*

Specific comments and edits (line numbers refer to the version with tracked changes):

L21: Clarify: interpolation -> spatial interpolation (or: interpolation to unmeasured areas)

*Changed to "spatial interpolation"*

L26: Since SAR penetration has become a large (and important) part of the manuscript, the introduction could be expanded with a paragraph on that topic using some of the material and references that are now in the data&methods section (L58-62, L89-91).

*Agree, we moved some of the more general descriptions of surface penetration and backscatter from the former methods section to the introduction (L.25-35): "… We use synthetic aperture radar (SAR) DEMs of the TanDEM-X satellites which are independent from cloud cover and provide a high spatial resolution. However, the SAR data derived elevation change rate can be biased by differences in signal penetration depth into the glacier surface between DEM acquisitions of different seasons. The depth of signal penetration is related to the prevailing glacier surface conditions at the acquisition time. In general, SAR penetration is close to zero for melting snow surfaces and bare glacier ice and increases during dry and frozen conditions. X-band penetration depths of several meters have been observed in*

*different regions (e.g. Millan et al., 2015; Zhao and Floricioiu, 2017; Abdullahi et al., 2018; Li et al., 2021). Previously, penetration depths have been estimated by a number of studies (e.g. Abdullahi et al., 2018, 2019; Li et al., 2021), using backscatter intensity. SAR backscatter intensity depends on physical properties of the glacier ice, such as grain size and density, roughness and water content (Wessel et al., 2016) and changes between melting and frozen conditions. Thus, we apply a regional correction for relative differences in SAR penetration, based on backscatter intensity, to account for different TanDEM-X acquisition periods of the Novaya Zemlya ice cap."*

Section 2.1: The main methods have now been moved to the supplement and only a brief summary remains here. I think this makes a much clearer distinction although it forces the reader to look up the supplement or the more methodology-oriented previous papers. The important density assumption should still be mentioned here with regards to the conversion from elevation change to mass change. And it would only take another sentence to also mention that you did separate elevation/mass rate calculations for land- and marine-terminating glaciers based on RGI. Also important for the Results/Discussion. And finally, it should be made clear that you make mass-change estimates both with and without the penetration correction (as in Table 1) and the associated terms should be clear and consistent to avoid confusion.

*We extended 2.1 and added explanations regarding the density conversion factor and land-/marine-terminating glaciers separation and corrected/uncorrected mass change calculations (L.119-125): "Glacier volume changes changes are converted to mass change using two density scenarios. For a) a conversion factor of 850±60 kg m$^{-3}$ (Huss, 2013) is applied and for b) 900±60 kg m$^{-3}$ as an approximation of the density of ice. For Novaya Zemlya, the geodetic mass change rate ($\Delta M/\Delta t$) is calculated using the uncorrected elevation change rate ($\Delta h/\Delta t_{uncorr.}$) as derived from the DEM differencing as well as the surface penetration corrected elevation change ($\Delta h/\Delta t_{corr.}$). Additionally, glacier elevation changes are derived specifically for marine- and land-terminating glaciers (Fig. S3), following the glacier terminus classification of the Randolph Glacier Inventory."*

*Concerning the calculation of mass changes with and without signal penetration correction, we also extended the supplementary methods (L. 87-97) to include the terminology.*

Section 2.2: This is a new subsection about X-band penetration. It reads fine, but is somewhat awkward in a data&methods section as it partly discusses SAR/X-band penetration in general (like introductory or discussion material) and partly describes the timing of the data without any further methodological description than "When comparing elevations of different TanDEM-X scenes the relative difference in penetration depths is determined by the acquisition dates and seasons", which really says nothing about how it was done. Instead, the penetration-related methods come in the next Section (2.3) which is narrowly named "SAR backscatter intensity analysis. I think it would be clearer to combine these two sections with a title that fits both, e.g. "Correction of seasonal penetration bias". Actually, since a penetration correction is added to the autumn DEMs before calculating mass change rates, I think it would be even better if this section comes first and describes both the TanDEM-X data and the correction, e.g. "2.1 TanDEM-X elevation data and penetration correction" and "2.2 Glacier elevation- and mass rates".

*Agree, we merged the former sections 2.2 & 2.3 (now section 2.1) and adjusted the internal structure of the methods chapter by moving the data (and penetration correction) description to the start of the chapter. The explanation of the elevation and mass change rate calculation is now section 2.2. In addition, some of the more general statements on surface penetration were moved to the introduction.*

*Also, the sentence "When comparing …" was rephrased to: *"When subtracting elevations of SAR DEMs from different seasons, the depth of signal penetration might differ between acquisitions, due to changing surface conditions, and bias the elevation change rate."*

Section 2.3: Nice add-on, but the topic is really penetration, not backscatter itself, so a revised title or a merge with Sect 2.2, as mentioned above, should be considered.

*See comment above*

Eqs. 1-2: Any references for these equations if they have been used in a similar context?

*Eq. 1 is just a trigonometric function based on the viewing geometry of the radar and the local surface slope. Eq. 2 is similar to the approach of (Abdullahi et al., 2019). However, we used a different version to derive the relative difference in penetration depths instead of absolute surface heights ((Abdullahi et al., 2019) applied a different linear model including SAR coherence). Therefore, we cited the studies in the methods section but not directly at the equation.

L122: smaller as -> smaller than

*Ok, changed

L148: add "average: ..." to the parentheses to be clear. What is "...confined to a smaller number of glaciers"? I think you mean that strong thinning is confined to a smaller number of glaciers, not the "general less negative" elevation changes.

*Yes, the sentence should express that strong thinning is only observed at a relatively small number of glaciers and not for all glaciers of the regions. Changed sentence (L.195-197) to: *"Regional average elevation changes of glaciers in Franz Josef Land ($-0.48\pm0.04$ m $a^{-1}$) and Severnaya Zemlya ($-0.34\pm0.12$ m $a^{-1}$) are in general less negative and strong thinning is confined to a smaller number of glaciers."*

L150: "Slight thickening is also observed...". What does "also" refer to? Delete.

*"also" referred to the description of positive elevation changes in the previous sentence (Vavilov surge). In any case, we removed it.

L152: The term "adjusted mass change" should have been clearly defined beforehand in the methods section, otherwise one has to guess what you mean here. Personally, I think "uncorrrected" and "corrected" would be clearer terms than "measured" and "adjusted", but either is fine as long as they are clearly defined at an early stage.

*We changed the terminology of the mass change results to "uncorrected mass change" *($\Delta M/\Delta t_{uncorr.}$)* and "corrected mass change" *($\Delta M/\Delta t_{corr.}$)* in the main manuscript and the supplementary methods. This new terminology is introduced in chapter 2.2 (glacier elevation change calculation)

L159: I understand what you mean, but the sentence is not technically clear. Rephrase.

*Split and rephrased sentence (L.180) to: *"The average vertical difference in surface elevation on the respective overlapping glacier areas (Fig. S2a) is 2.13 m. Also, the elevation change rates derived from all glacier areas which were acquired in September 2016 (Fig. 1b), show an average difference in surface lowering of 0.4 m a⁻¹ compared to areas acquired in winter 2016/17."*

L156-168: I suggest to move this text about the penetration issue ahead of the glaciological results, similar to what I suggested for the data&methods section and what is already done in the Discussion. First you build confidence in the data and corrections, then you look at glacier changes.

*Agree, we changed the structure of the results chapter accordingly.

L194: What is "relatively large"? Based on your calculations and relevant studies, you can provide an approximate meter-range here to make it more informative.

*We did not provide a number in this sentence deliberately because our estimate shows only a relative difference in surface penetration between winter and autumn and not an absolute depth of penetration. The latter can not be derived from our datasets as we do not have a height reference for the respective acquisition dates. Thus, it might confuse readers to provide an absolute depth of penetration at this point which cannot inferred from our TanDEM-X data. However, based on other studies which did calculate absolute penetration depths, it is likely that the winter penetration was in the range of several meters. Therefore, we extended the sentence (L.222): *"... was relatively large, i.e. several meters as found by previous studies (Millan et al., 2015; Zhao and Floricioiu, 2017; Abdullahi et al., 2018; Li et al., 2021), ..."*

L195: An important reference study, but it is not really a "similar observation" because your relative penetration differences indicate penetration beyond previous summer surface which is not expected to be deeper than 1-2 m in your case. If the summer-surface indeed plays a role here, it can be troublesome because there are large variations from year-to-year in how much melt and refreezing there is in the higher accumulation area. Some years almost no melt and refreezing, other years thick ice layers can form. I miss a brief discussion of this issue in relation to the findings of Rott et al.

*Agree, for the glacier areas with relatively high differences in penetration depth, the scattering surface of the winter acquisitions does probably not refer to the late summer surface of the previous year. For those areas with high penetration values the measured height of the phase center could be either related to volume scattering into the ice body or an even older firn layer. This could be the case, for example, for one of the DEM strips in the northern part of Novaya Zemlya (Fig. S2b) which shows high offset values in the upper accumulation area of the ice cap. However, without knowledge of the glacier ice stratification it is not possible to determine the potential cause. On the other side, there are also glacier areas which show small differences in comparison. For those areas, it is not unlikely that the microwave signal is reflected by the summer surface of the previous year.

We discuss this in the revised manuscript (L.223-230)*: "For TanDEM-X DEMs of the Antarctic Peninsula it was observed that the measured cold-season heights rather referred to the refrozen firn of the previous summer than to the actual glacier surface (Rott et al., 2014). This might be also the case for some of the glacierized areas of Novaya Zemlya where the observed bias between autumn and winter DEMs is relatively small (e.g. < 2 m). However, for glacier areas with higher differences in signal penetration depth, it is more likely that the measurement is biased by penetration beyond the previous late-summer surface, either by an older ice layer of a year with widespread melt and refreezing or volume scattering of the X-band SAR* (Dall et al., 2001). "

L221: More negative than what? Franz Josef Land or the other studies. Unclear sentence.

*This referred to the gravimetric measurements mentioned at the beginning of the sentence. Changed to (L.250): *"..., the estimate for Severnaya Zemlya is even more negative than the gravimetric measurements (Ciracì et al., 2020) which might indicate recent acceleration of glacier mass loss also on this archipelago."*

L224: Sudden transition from discussing mass changes to elevation changes. Please reread the Discussion and try to be more consistent with terms and quantities discussed.

*For the glacier change discussion, there are a variety of studies with different regional settings and methodology. We decided to use a structure which begins with the large-scale regional studies (GRACE & Aster), followed by region-specific studies which focus on the individual archipelagos. Thereby, it is not entirely possible to avoid transitions between different datasets and methods. Nevertheless, we rephrased some parts of the discussion and reduced the transitions between different units. In the former L.224, we replaced the elevation change value with the respective mass change value.

L224: Marine-terminating only or land-terminating also? Fig. S3 indicate little difference between the two types at NZ, which I think should have been discussed too. But here you maybe talk about a smaller number of glacier with even larger changes, please specify.

*The strongest negative elevation changes are measured at the termini of some marine-terminating glaciers on Novaya Zemlya (particularly northwestern part of the ice cap). However, there are also a number of marine-terminating glaciers with less negative surface changes which is the reason why the average changes in Fig.S3 are not very different for marine- and land-terminating glaciers. Nevertheless, this statement referred to the (marine-terminating) outlet glaciers of Novaya Zemlya which have the most negative elevation changes. Thus, we rephrased the sentence (L.249)*: "The strongest local surface lowering is observed at some of the large marine-terminating outlet glaciers, most notably on Novaya Zemlya (Northwestern Severny Island ice cap)."*

L229: If terms have been well explained earlier, you wouldn't need a long add-on like "...to the signal penetration adjusted mass change rate derived by TanDEM-X".

*Agree, we changed the mass change terminology (see comment 3) and shortened this part.

L230: "also measured by Strozzi". What does "also" refer to? Your results show few indications of accelerated flow on NZ as far as I can see, so it's unclear how your results relate to those of Strozzi et al. Please clarify the relevance in the manuscript.

*This phrase is more related to the previously cited studies (Carr et al., 2014; Melkonian et al., 2016) which describe an increasing retreat of large outlet glaciers and general increase in mass loss on Novaya Zemlya during (approximately) similar observation periods as the TanDEM-X difference of this study. To clarify this, we moved the sentence to Line 255: *"... For those glaciers, an increasing retreat in the early 21st century was attributed to fjord geometries and changes in sea-ice concentrations (Carr et al., 2014). Long-term observations indicate a more rapid thinning during recent years, particularly at the termini of marine-terminating glaciers (Melkonian et al., 2016). An acceleration in flow velocities for*

*the major tidewater glaciers in the Russian Arctic was also measured by (Strozzi et al., 2017) over the course of the last decades"*

L233: "does not seem to be related to potential SAR penetration" -> "do not seem to be related to differences in SAR penetration" (there is penetration, but it can vary...)

*Agree, changed sentence accordingly

L236: Or it can be related to a long-term dynamic imbalance (too low ice-flux velocities) with cyclic fast-flow/surging as seen for some glaciers. Your results are not conclusive.

*Yes, it is not possible to conclude this from our rather short observation period. We therefore wrote *"... might be related ..."* because the elevation gains cannot be clearly attributed to increases in moisture.

L241-242: This is your conclusion based on a single observation period. The term "increasingly negative" infers a more continuous process (acceleration) which you cannot conclude from your data and is not supported from gravimetry time series either. To be on the safe side, you should limit your conclusion to an increased mass loss in your period versus other studies from the 2000s. Please keep this in mind elsewhere too.

*This sentence was intended to describe the increase in mass loss between our study and measurements between ~2000 – 2010. Also, the most recent gravimetric time series (Ciracì et al., 2020) does indicate an acceleration for FJL & NZ (their Table 1). Nevertheless, we made some small changes to clarify this comparison (L.276-280): *"Glaciers in the Russian High Arctic have shown a contribution of 0.06 mm a$^{-1}$ to global sea-level rise between 2010 and 2017 and an increased mass loss compared to the first decade of the 21$^{st}$ century."*

L243: As pointed out in the previous review, this sentence is unclear. Please revise according to the explanation in the author response letter which makes sense.

*Extended/changed conclusion (L.278-280): *"This observation is in line with glacier changes of other Arctic regions, showing an increasing contribution to sea-level rise in the last decades. While specific mass change rates of Arctic glaciers are still less negative than those of many glaciers outside the polar regions, the absolute mass loss is higher due to the vast glacierized areas of the Arctic."*

Fig 1: Nice figure, but the data coverage of each panel should be made clear by statements for panel a and b (all data or only overlapping?) and by referring to Fig. S2 for aerial extents. This important aspect can also be made clearer in the manuscript text.

*The data points shown in Fig. 1a refer to a random subset (for improved visibility) of all raster cells of the 2016/17 DEM mosaic. The hypsometric mean elevation changes in Fig 1b were also calculated from all respective glacier areas (not only overlapping). Fig. 1c & d refer to overlapping glacier areas. We extended the caption to clarify the data shown in each panel: Fig. 1a: *"Point icons illustrate a random subset (5000 cells) of the 2016/17 DEM mosaic of Novaya Zemlya."* & Fig. 1b: *"Mean elevation change rates ... of all respective glacier areas."*

Fig. 2, caption: Nice figure. To be more precise I would say: "...corrected for differences in seasonal SAR-signal penetration (Fig. S2)."

*Ok, changed caption accordingly

Supp-L55: How was the co-registration done?

*The co-registration includes an initial vertical correction, followed by an iterative horizontal correction (following the algorithm of (Nuth and Kääb, 2011)) and a final vertical correction. The description of the co-registration was extended (L.57-61): *"In both cases, the co-registration is performed as an iterative process to remove vertical and horizontal shifts between the "raw" CoSSC DEMs and a reference surface. Initially, vertical biases are estimated (on stable areas) and corrected. Thereafter, horizontal shifts are minimized using an iterative approach of (Nuth and Kääb, 2011). Eventually, a second vertical correction is applied to reduce remaining offsets."*

Fig. S2b: Penetration-bias corrections look very different from scene to scene, which I suppose is due to different backscattering conditions for the acquisitions. Although this can be a large local issue, it has only a small impact on the regional change rates which is the main scope of this paper. Still, a brief discussion of this matter would be good.

*Yes, the visible differences in the correction field are related to the different acquisition dates of the September TanDEM-X data and reflect the varying states of backscatter intensity and local surface conditions. Due to the limited extent of overlapping data takes, the proposed correction scheme can only account for mean differences between autumn/winter acquisitions. It is therefore possible that on a local scale the applied correction, as mentioned, over- or underestimates the actual signal penetration depth.

*We added this to the discussion in the main manuscript: *"It is noteworthy, that the applied regional correction scheme can introduce a larger uncertainty at a local glacier scale caused by different surface and backscatter conditions between the specific TanDEM-X acquisitions (Fig. S2b). However, due to the limited extent of overlapping glacier areas (Fig. S2a), it is not possible to derive a date-specific intensity correction for each DEM strip. Thus, the applied linear model does rather represent an average difference in surface penetration depth between autumn and winter SAR data."*

*… and also to the caption of Fig. S2: *"Vertical differences in the estimated correction field are caused by different local surface conditions and backscatter intensities of each September TanDEM-X acquisition."*

**Author´s response: Increased glacier mass loss in the Russian High Arctic (2010-2017)**

*References:*

Abdullahi, S., Wessel, B., Leichtle, T., Huber, M., Wohlfart, C., and Roth, A.: Investigation of Tandem-x Penetration Depth Over the Greenland Ice Sheet, in: IGARSS 2018 - 2018 IEEE International Geoscience and Remote Sensing Symposium, IGARSS 2018 - 2018 IEEE International Geoscience and Remote Sensing Symposium, Valencia, 1336–1339, https://doi.org/10.1109/IGARSS.2018.8518930, 2018.

Abdullahi, S., Wessel, B., Huber, M., Wendleder, A., Roth, A., and Kuenzer, C.: Estimating Penetration-Related X-Band InSAR Elevation Bias: A Study over the Greenland Ice Sheet, 19, 2019.

Carr, J. R., Stokes, C., and Vieli, A.: Recent retreat of major outlet glaciers on Novaya Zemlya, Russian Arctic, influenced by fjord geometry and sea-ice conditions, J. Glaciol., 60, 155–170, https://doi.org/10.3189/2014JoG13J122, 2014.

Ciracì, E., Velicogna, I., and Swenson, S.: Continuity of the Mass Loss of the World's Glaciers and Ice Caps From the GRACE and GRACE Follow-On Missions, Geophys. Res. Lett., 47, https://doi.org/10.1029/2019GL086926, 2020.

Dall, J., Madsen, S. N., Keller, K., and Forsberg, R.: Topography and penetration of the Greenland Ice Sheet measured with Airborne SAR Interferometry, Geophys. Res. Lett., 28, 1703–1706, https://doi.org/10.1029/2000GL011787, 2001.

Huss, M.: Density assumptions for converting geodetic glacier volume change to mass change, The Cryosphere, 7, 877–887, https://doi.org/10.5194/tc-7-877-2013, 2013.

Li, J., Li, Z.-W., Hu, J., Wu, L.-X., Li, X., Guo, L., Liu, Z., Miao, Z.-L., Wang, W., and Chen, J.-L.: Investigating the bias of TanDEM-X digital elevation models of glaciers on the Tibetan Plateau: impacting factors and potential effects on geodetic mass-balance measurements, J. Glaciol., 1–14, https://doi.org/10.1017/jog.2021.15, 2021.

Melkonian, A. K., Willis, M. J., Pritchard, M. E., and Stewart, A. J.: Recent changes in glacier velocities and thinning at Novaya Zemlya, Remote Sens. Environ., 174, 244–257, https://doi.org/10.1016/j.rse.2015.11.001, 2016.

Millan, R., Dehecq, A., Trouve, E., Gourmelen, N., and Berthier, E.: Elevation changes and X-band ice and snow penetration inferred from TanDEM-X data of the Mont-Blanc area, in: 2015 8th International Workshop on the Analysis of Multitemporal Remote Sensing Images (Multi-Temp), 2015 8th International Workshop on the Analysis of Multitemporal Remote Sensing Images (Multi-Temp), Annecy, France, 1–4, https://doi.org/10.1109/Multi-Temp.2015.7245753, 2015.

Nuth, C. and Kääb, A.: Co-registration and bias corrections of satellite elevation data sets for quantifying glacier thickness change, The Cryosphere, 5, 271–290, https://doi.org/10.5194/tc-5-271-2011, 2011.

Rott, H., Floricioiu, D., Wuite, J., Scheiblauer, S., Nagler, T., and Kern, M.: Mass changes of outlet glaciers along the Nordensjköld Coast, northern Antarctic Peninsula, based on TanDEM-X satellite measurements: TanDEM-X Antarctic Peninsula glaciers, Geophys. Res. Lett., 41, 8123–8129, https://doi.org/10.1002/2014GL061613, 2014.

Strozzi, T., Paul, F., Wiesmann, A., Schellenberger, T., and Kääb, A.: Circum-Arctic Changes in the Flow of Glaciers and Ice Caps from Satellite SAR Data between the 1990s and 2017, Remote Sens., 9, 947, https://doi.org/10.3390/rs9090947, 2017.

Wessel, B., Bertram, A., Gruber, A., Bemm, S., and Dech, S.: A NEW HIGH-RESOLUTION ELEVATION MODEL OF GREENLAND DERIVED FROM TANDEM-X, ISPRS Ann. Photogramm. Remote Sens. Spat. Inf. Sci., III–7, 9–16, https://doi.org/10.5194/isprsannals-III-7-9-2016, 2016.

Zhao, J. and Floricioiu, D.: THE PENETRATION EFFECTS ON TANDEM-X ELEVATION USING THE GNSS AND LASER ALTIMETRY MEASUREMENTS IN ANTARCTICA, ISPRS - Int. Arch. Photogramm. Remote Sens. Spat. Inf. Sci., XLII-2/W7, 1593–1600, https://doi.org/10.5194/isprs-archives-XLII-2-W7-1593-2017, 2017.

**Response to reviewer 2:**

2nd report for the Brief communication paper: **Increased glacier mass loss in the Russian Arctic (2010-2017)**

The Cryosphere Discuss. https://tc.copernicus.org/preprints/tc-2020-358/

This 2nd report refers to the revised manuscript **tc-2020-358-manuscript-version3.pdf** and **tc-2020-358-supplement-version3.pdf** from 26.05.2021

**Comments to the authors**

Thank you for responding to my comments and the changes implemented for improving the work. The paper is better structured now the Data and Methods section reveals relevant aspects specific to the used dataset on this particular glaciated region. Although, as suggested in the review of the first version, the analysis of the backscattering coefficients was added, the estimation of the penetration depth of the X-band SAR signal into the glacier volume is based on wrong assumptions.
*Thank you very much for your comments and please find our point-by-point responses below. Line numbers refer to the track-changes version of the revised manuscript.*

I have some doubts regarding the correctness of Eq.1 by the following reasons (see also Dall, 2007): (i) The penetration depth (dp) refers to the vertical. (ii) For small relative penetration the elevation bias hb can be approximated by the two-way power penetration depth: $dp_2 = dp/2 \approx h_b$. (not by the one-way penetration depth). (iii) For given InSAR geometry and propagation conditions (permittivity) dp is related to the oblique radar propagation path multiplied by the cos of the refraction angle in the snow volume.
*As suggested, we applied the approach using the two-way power penetration to estimate the surface penetration depth instead of the trigonometric function. To estimate the refraction angle into the glacier surface, we referred to a reference study on in-situ experiments in Antarctica (see below). Using this approach, the following paragraphs would replace the former Eq. 1 (L.70) in the revised manuscript):*

*"The vertical differences between heights of autumn and winter DEM acquisitions are converted into depths of signal penetration into the glacier volume using Eq. 1 following (Dall, 2007):*

$$l = \frac{d_p}{\cos(\Theta_v)} \; ; \; d_p = 2 \times h_b \qquad \text{Eq. 1}$$

*where $l$ is the penetration length and $\Theta_v$ the refraction angle into the volume. $d_p$ is the two-way power penetration depth and can be approximated by two times the vertical elevation bias $h_b$ (Dall, 2007). To derive the refraction angle ($\Theta_v$), Eq. 2 (Snell´s law) is applied:*

$$\sin(\Theta_v) = n_1 \times \frac{\sin(\Theta_v)}{n_2} \qquad \text{Eq. 2}$$

*where $\Theta_l$ is the local incidence angle, $n_1$ the refractive index of air (1.000293) and $n_2$ the refractive index of glacier ice. For the permittivity of ice, various values have been reported in literature (Rasmussen, 1986; Dowdeswell and Evans, 2004). In general, the refractive index of ice increases with depths due to changes in density. Therefore, we refer to a detailed in-situ study on refraction measurements from the*

ice surface down to depths of 150m in Antarctica (Kravchenko et al., 2004). For glacier ice close to the surface (0 to -40 m depth), they found values between ~1.3 and ~1.5 as index of refraction. Thus, we apply a refractive index of ice ($n_2$) of 1.4 as the approximate permittivity of ice close to the glacier surface."

We also recalculated the signal penetration corrected elevation & mass change for Novaya Zemlya with this approach but the results are almost exactly the same as in the original version (original $\Delta h/\Delta t$ = -0.643 m/a and new $\Delta h/\Delta t$ = -0.644 m/a). The only significant change would be the estimated average vertical offset (3.5 m instead of 2.13 m).

However, we are not sure if this approach improves the accuracy of the estimate. The model presented by (Dall, 2007) assumes an infinite volume, i.e. the microwave signal is not scattered by any layer below the surface (volume scattering). While this might be the case for some glacier areas on Novaya Zemlya with rather high vertical offsets, it is not unlikely that there is a scattering layer below the actual surface (e.g. melt/refrezzing of a previous summer) for glacier areas with smaller vertical differences. For those areas, the new approach could overestimate the penetration depth and produce a rather high average penetration depth for all September acquisitions.

For those reasons, we did not include the two-way power penetration estimate into the revised manuscript yet, but we would be very interested to hear the reviewer´s opinion regarding those concerns.

The penetration bias depends not only on the radar wave propagation properties in the snow volume but also on the interferometric baseline and incidence angle. The impact of these parameters needs to be considered if an observed elevation bias (or penetration value) is applied to another InSAR scene.

*We did not correct for different incidence angles or baselines because the viewing geometries of the used SAR data on Novaya Zemlya are relatively similar: 99% of the autumn glacier areas have been acquired with a difference in incidence angles of less than 2°. More precisely, 68% of the raster cells have an incidence angle of ~39.3° and the remaining 32% of ~41.3°. Regarding the baseline, the mean effective baseline of ~93% of the used data is 91.9 m with a minimum to maximum range between 87.8 and 95.4 m. Only the acquisitions of 2016-09-08 (~7% of total area) have a significantly larger baseline of 238.6 m. In the revised version, the specific incidence angles and baselines of each DEM strip are stated in Table S2 and we added the following lines to the methods section (L. 88-91): *"We did not adjust for differences in incidence angle or effective baseline because the viewing geometries of the majority of the used SAR acquisitions are rather similar (Table S2). For 99% of the glacierized area of Novaya Zemlya, the difference in incidence angles is not larger than 2° (39.3° - 41.3°) while for 93% of area the average baseline is 91.9 m (87.8 m – 95.4 m)."*

Regarding the Fig. 1 and the related text (line 70 and below):

For estimating the penetration-related elevation bias in Eq. 2 the difference in σ0 between September (surface melt) and mean σ0 of Oct. to Jan. is used as proxy. This implies an immediate switch for melting state in Sept. to dry snow with deep penetration in Oct. In reality this transition is gradual in time which means using October (and possibly also November) data in the "winter" ensemble causes a bias for estimating the penetration for the winter case. In Fig. 1a the Oct. and Nov. σ0 values are lower than the Jan. values (in particular in the 300 m to 600 m elevation zone).

*Agree, a gradual transition between melting and frozen conditions is much more realistic than a sudden transition. However, the extent of overlapping glacier areas (which are used to estimate the vertical offset) is limiting the potential of a more detailed comparison. For example, if we would calculate a specific correction factor for each month (or even acquisition date), the reference areas would be very small. Additionally, another source of uncertainty could be the spatial distribution of reference areas across the Novaya Zemlya ice cap. In the current comparison, the overlapping (reference) areas are almost equally distributed across Novaya Zemlya (Fig. S2). When using a monthly or sub-monthly comparison, the individual reference areas would be no longer represent the entire latitudinal extent of

Novaya Zemlya. Therefore, we think that a correction approach with a higher temporal resolution would rather increase the uncertainty than improve the results. Nevertheless, we agree that this average correction factor can over- or underestimate differences in signal penetration depth on a local (glacier-specific) scale, depending on the exact acquisition times. To discuss this issue, we added a paragraph to the discussion (L.235): *"It is noteworthy, that the applied regional correction scheme can introduce a larger uncertainty at a local glacier scale caused by different surface and backscatter conditions between the specific TanDEM-X acquisitions (Fig. S2b). However, due to the limited extent of overlapping glacier areas (Fig. S2a), it is not possible to derive a date-specific intensity correction for each DEM strip. Thus, the applied linear model does rather represent an average difference in surface penetration depth between autumn and winter SAR data."*

Fig. 1a: The used procedure (calibration coefficients) to convert amplitude to σ0 needs to be checked because as far as I see σ0 values are down to -30 dB which is far below NESZ. I also miss mentioning in the paper or supplement in which way was the incidence angle dependence of backscatter intensity taken into account. Also, the look angle of the various TanDEM-X acquisitions is not given anywhere (Table S2 gives a list but with some redundant information). In particular for wet snow σ0 show large changes with the incidence angle.

*The intensity images are created using the Gamma remote sensing software environment (Werner et al., 2000). The radiometric calibration of the amplitude to σ0 values is automatically performed by the conversion algorithm from the CoSSC to the Gamma data format (using the metadata of the CoSSC data product). The respective algorithms are part of the interferometry (ISP) module and described in the *Interferometric SAR Processor – ISP user´s guide* (GAMMA Interferometric SAR Processor (ISP), 2021) in section 2.2.7 (TerraSAR-X & TanDEM-X data read algorithms) and 2.4.5 (radiometric calibration procedure). A link to this user guide is provided in the reference list.

*Concerning the incidence angle, we revised Table S2 following the suggestions (see respective comment below) and extended the methods section (see second comment).

Results (line 92 and below and Fig. 2)

The error bar is decreasing with the decreasing magnitude of Δh/Δt and increasing elevation. Usually, the geodetic error should be independent on Δh/Δt. At higher elevations where Δh/Δt small additional error contributions may be added resulting in larger error bars than at the termini. I recommend therefore to revise the error calculations. Regarding the uncertainty assessment for the mass change (now equation (1) in Supplement in the current version of the manuscript) I also have some doubts (expressed also by reviewer #2). According to this equation the error of the mass change estimate depends on the mass change magnitude ΔM/Δt. This would mean a zero mass change estimate would yield a perfect result (no error). But then the first term of the sum would compensate: small (near zero) Δh/Δt leads to very large error and vice versa (in case of large mass changes). These terms contributing to the error budget should be treated independently to hold for quadrature sum. See also (Nuth & Kääb, 2011).

*The hypsometric bars shown in Fig. 2 refer to the normalized median absolute deviation of Δh/Δt measurements on glacier areas within each elevation bin. Therefore, the bars are largest a low elevations because the spread of measured Δh/Δt values is large due to the presence of strong thinning glacier termini. At high altitudes, the range of measured Δh/Δt values is in general much smaller (see also Δh/Δt maps of Fig.2) and thereby also the bar. We extended the caption of Fig. 2 because the description of the shown bars was missing. Regarding the geodetic error, we did not calculate a mass change error for each elevation bin but for the entire region (based on the mean regional elevation change and respective uncertainty).

*Regarding to Eq. 1 (Supplement), unfortunately we do not quite understand the question referring to the mass change and elevation change uncertainty: The first term of the sum is the ratio between the Δh/Δt uncertainty ($\delta_{\Delta h/\Delta t}$) and the mean (glacier) Δh/Δt estimate. While the Δh/Δt estimate is derived on glacierized areas, the Δh/Δt uncertainty ($\delta_{\Delta h/\Delta t}$) mainly indicates the potentially remaining offsets on non-glacier areas after the co-registration (and also other sources of uncertainty, Supplement Eq.2). If Δh/Δt

would be very small (and thereby also ΔM/Δt), the uncertainty of ΔM/Δt could still be relatively high if $\delta_{\Delta h/\Delta t}$ (off-ice) is high compared to Δh/Δt (on-ice). For example, the measured elevation change rate is rather small but there are a lot of artificial elevation offsets remaining after the co-registration. In this case $\delta_{\Delta M/\Delta t}$ would be high compared to ΔM/Δt.

**Specific comments**

Main paper:

Line 39 into the glacier volume.
*Included "volume"

Line 41 increases in dry snow.
*Ok, changed

Line 51 snow and ice properties at the glacier surface can have significant impact on …
*Included "properties"

Line 58 much lower backscatter values then …
*Ok, corrected

Line 64 and 66 replace "surface penetration" by penetration into the volume
*Ok, replaced

Line 86 smaller than on NZ
*Ok, corrected sentence

Line 89 Fig S1a
*Ok, changed

Line 298 Fig. 1a (identical with Fig S1e): Mean backscatter corresponding to 2016-12 is not visible
*Fig. 1a shows a subset of backscatter values (5000 random samples) because otherwise the figure would be too busy. The acquisitions of December 2016 cover only a very small fraction of the Novaya Zemlya ice cap (~10 km²). For this reason, there are only very few December datapoints visible and the mean backscatter was only calculated and plotted for the lowest elevation bin. The mean value is almost the same as for October 2016 (triangle) and therefore difficult to identify in the figure.

Supplement

Adding Table S2 is welcome but contains a lot of redundant information and not the important one. One row pro TanDEM-X acquisition (instead of one row pro CoSSC framing of the same datatake) would be enough but some additional information would be useful: Beff, HoA, incidence angle, etc similar to other publications using self-processed TanDEM-X DEMs (e.g. Table 1 in (Malz et al., 2018)). Keep the established acronyms and labels used in the metadata: Active sensor instead of "transmitting", "Strip" should be "Beam" and TSX-1 and TDX-1 (instead of TST and TDT), Relative orbit instead of "Path number".
*We changed Table S2 and included only one row per TanDEM-X acquisition instead of each CoSSC frame. The terminology was adjusted and the new columns include now, as suggested, acquisition date, acquisition start time, active satellite, orbit direction, relative orbit, strip length (number of CoSSC frames), effective baseline, height of ambiguity and incidence angle.

Line 122 quadrature sum
*Ok, changed

**References:**

Dall, J.: InSAR elevation bias caused by penetration into uniform volumes, IEEE Trans. Geosci. Remote Sens., 45, 2319–2324, 2007

Malz, P.; Meier, W.; Casassa, G.; Jaña, R.; Skvarca, P.; Braun, M.H. Elevation and Mass Changes of the Southern Patagonia Icefield Derived from TanDEM-X and SRTM Data. *Remote Sens.* **2018**, *10*, 188. https://doi.org/10.3390/rs10020188

Nuth, C. and Kääb, A.: Co-registration and bias corrections of satellite elevation data sets for quantifying glacier thickness change, The Cryosphere, 5, 271–290, https://doi.org/10.5194/tc-5-271-2011, 2011.

*References*

Dall, J.: InSAR Elevation Bias Caused by Penetration Into Uniform Volumes, IEEE Trans. Geosci. Remote Sens., 45, 2319–2324, https://doi.org/10.1109/TGRS.2007.896613, 2007.

Dowdeswell, J. A. and Evans, S.: Investigations of the form and flow of ice sheets and glaciers using radio-echo sounding, Rep. Prog. Phys., 67, 1821–1861, https://doi.org/10.1088/0034-4885/67/10/R03, 2004.

Kravchenko, I., Besson, D., and Meyers, J.: In situ index-of-refraction measurements of the South Polar firn with the RICE detector, J. Glaciol., 50, 522–532, https://doi.org/10.3189/172756504781829800, 2004.

Rasmussen, L. A.: REFRACTION CORRECTION FOR RADIO ECHO-SOUNDING OF ICE OVERLAIN BY FIRN, J. Glaciol., Vol. 32, 192–194, 1986.

GAMMA Interferometric SAR Processor (ISP): https://esdynamics.geo.uni-tuebingen.de/wiki/files/remote_sensing/pdf/ISP_users_guide.pdf, last access: 6 September 2021.

Werner, C., Wegmüller, U., Strozzi, T., and Wiesmann, A.: GAMMA SAR AND INTERFEROMETRIC PROCESSING SOFTWARE, 9, 2000.

---

## Author Response (AR3)

**Response to 3rd report for the Brief communication paper: Increased glacier mass loss in the Russian Arctic (2010-2017)**

The Cryosphere Discuss. https://tc.copernicus.org/preprints/tc-2020-358/

This 3rd report refers to the revised manuscript tc-2020-358-manuscript-version4.pdf and tc-2020-358-supplement-version4.pdf from 21.09.2021

Green: Response of the authors to reviewer 2

Blue: Report #3

Bold black: Author responses to report #3 of reviewer 2

**Comments to the authors**

Thank you for responding one more time to my comments and the changes implemented for improving the work. I address here the critical issues remaining to be clarified.

• Thank you very much the comments and please find our point-by-point responses below. Regarding the conversion between observed vertical elevation differences and signal penetration length, we suggest to include the original version as well as the two-way power penetration approach in the supplement (new sections 3.1 & 3.2) because both approaches are based on assumptions (e.g. volume scattering or the presence of a scattering late-summer firn layer) which might be correct for some but not all glacier areas of Novaya Zemlya. Also, both methods produce almost the same vertical correction fields. By this means, we also save some space in the methods section of the main manuscript.

\*As suggested, we applied the approach using the two-way power penetration to estimate the surface penetration depth instead of the trigonometric function. To estimate the refraction angle into the glacier surface, we referred to a reference study on in-situ experiments in Antarctica (see below). Using this approach, the following paragraphs would replace the former Eq. 1 (L.70) in the revised manuscript):

"The vertical differences between heights of autumn and winter DEM acquisitions are converted into depths of signal penetration into the glacier volume using Eq. 1 following (Dall, 2007):

$$l = \frac{d_p}{\cos(\theta_v)}$$
;  $d_p = 2 \times h_b$  Eq. 1

where I is the penetration length and  $\Theta v$  the refraction angle into the volume. dp is the two-way power penetration depth and can be approximated by two times the vertical elevation bias hb (Dall, 2007). To derive the refraction angle ( $\Theta v$ ), Eq. 2 (Snell's law) is applied:

$$\sin(\Theta_{\nu}) = n_1 \times \frac{\sin(\Theta_l)}{n_2}$$
 Eq. 2

where OI is the local incidence angle, n1 the refractive index of air (1.000293) and n2 the refractive index of glacier ice. For the permittivity of ice, various values have been reported in literature (Rasmussen, 1986; Dowdeswell and Evans, 2004). In general, the refractive index of ice increases with depths due to changes in density. Therefore, we refer to a detailed in-situ study on refraction measurements from the ice surface down to depths of 150m in Antarctica (Kravchenko et al., 2004). For glacier ice close to the surface (0 to -40 m depth), they found values between ~1.3 and ~1.5 as index of refraction. Thus, we apply a refractive index of ice (n2) of 1.4 as the approximate permittivity of ice close to the glacier surface."

I would have preferred a revision in the manuscript directly to avoid confusion. The change of Eq.1 after (Dall, 2007) is welcome. In this paper the vertical penetration bias (hb) is approximately the twoway power penetration depth (dp2) in the case of a small penetration compared to the height of ambiguity (eq. 13 in (Dall, 2007)). With dp the one-way power penetration depth is denoted.

• Agree, there was a typo in the equation. We changed it to:  $l_p = \frac{d_{p2} \times 2}{\cos(\theta_r)}$ ;  $d_{p2} \approx h_b$  Eq. 2

But since you already have an observed height difference I suggest to derive the correction directly.

The linear correction could be derived directly from the observed height differences without
the conversion into signal penetration length. Also, the overall results of the correction with
and without the conversion are relatively similar. However, in our understanding, by removing the conversion (based on the local incidence and surface slope) this approach would neglect the local topography as the glacier surface is not a flat area. Thus, we think that the
conversion between observed vertical differences and signal penetration lengths should be
included.

In the manuscript dp is once defined as depth of penetration into the volume (line 61) and below (line 68) as penetration bias. These are not the same. In this case I guess EQ2 refers to the penetration related elevation bias hb.

• Unfortunately, there was some confusion between the linear regression equation (EQ2 in the paper) and the response letter. EQ2 is applied to estimate the penetration length into the glacier volume (not vertical elevation difference). We decided to use this variable because we wanted to account for the surface topography as the local surface slope and incidence angle is known for all glacier areas. In the original EQ1 of the previous manuscript version *dp* was not defined as the vertical difference (in contrast to (Dall, 2007)). Therefore, in the new version *dp* in Eq.2 was replaced with *lp* which is the penetration length into the volume instead of the vertical difference.

**In Author's response Eq.2 right: sin $\Theta_1$**

**But you don't need to apply Snell's law if the penetration is defined as vertical. If yes, please explain.**

• The estimated penetration in the regression model is not defined as vertical (see comment above). Therefore, Snell's law has to be applied, in our understanding, to account for the slight change in direction of the X-Band signal at the interface (glacier surface) between atmosphere and glacier ice.

Besides Kravchenko et al. report on measurements of the dielectric permittivity on the South Pole, with a density profile of cold polar firn, very different from that of the percolation zone of Arctic glaciers. The real part of permittivity of dry snow and firn can be computed from the density with a slightly non-linear relation (Ulaby and Long, 2014). Typical density profiles from the percolation zone of Artic glaciers have been reported in several papers (e.g. for Svalbard by Marchenko et al., 2017).

For the dielectric permittivity of the glacier ice close to the surface, we had to refer to values from the literature as there are no density profiles available from the Russian Arctic archipelagos (to our knowledge). The values reported by (Kravchenko et al., 2004) are in fact from measurements in Antarctica but they also refer to a low density (~400 kg m.3) for the surface layers which is similar to the surface part of the profiles shown by (Marchenko et al., 2017) (their figure 4). Therefore, we assume that the used value is a reasonable approximation for our study region.

We also recalculated the signal penetration corrected elevation & mass change for Novaya Zemlya with this approach but the results are almost exactly the same as in the original version (original  $\Delta h/\Delta t$  = -0.643 m/a and new  $\Delta h/\Delta t$  = -0.644 m/a). The only significant change would be the estimated average vertical offset (3.5 m instead of 2.13 m).

However, we are not sure if this approach improves the accuracy of the estimate. The model presented by (Dall, 2007) assumes an infinite volume, i.e. the microwave signal is not scattered by any layer below the surface (volume scattering). While this might be the case for some glacier areas on Novaya Zemlya with rather high vertical offsets, it is not unlikely that there is a scattering layer below the actual surface (e.g. melt/refrezzing of a previous summer) for glacier areas with smaller vertical differences. For those areas, the new approach could overestimate the penetration depth and produce a rather high average penetration depth for all September acquisitions.

For those reasons, we did not include the two-way power penetration estimate into the revised manuscript yet, but we would be very interested to hear the reviewer's opinion regarding those concerns.

EQ2 (line 70) should refer to the  $\Delta$ hW-A.(or hb) A plot of its altitude dependence over the overlapping areas (red dots in Figure S2a) would be crucial to understand how this regression was derived. The regressions shown in Fig 1c and Fig 1d are not explained at all. Hard to understand how they contribute to EQ2.

- EQ2 is used to estimate the penetration length which is then converted back to the vertical elevation bias using the respective equations (also see comments above). There was unfortunately some confusion because *dp* was defined vertical in the response letter but not in the original manuscript.
- Agree, we added another panel to Figure S2 which shows the average observed vertical difference of the indicated overlapping glacier areas.
- The linear regressions shown in Fig 1c & 1d do not contribute directly to the correction function. We included those regression lines in the figures to better illustrate the correlations between backscatter intensity, elevation and differences in signal penetration depth. We mentioned those connections in section 2.1 and extended the caption of Figure 1: "The linear correlations of mean September backscatter intensity and elevation (1c) and mean difference in signal penetration depth and September backscatter intensity (1d) are indicated as black solid lines."

"We did not adjust for differences in incidence angle or effective baseline because the viewing geometries of the majority of the used SAR acquisitions are rather similar (Table S2). For 99% of the glacierized area of Novaya Zemlya, the difference in incidence angles is not larger than  $2^{\circ}$  (39.3° - 41.3°) while for 93% of area the average baseline is 91.9 m (87.8 m - 95.4 m)."

Accepted.

"It is noteworthy, that the applied regional correction scheme can introduce a larger uncertainty at a local glacier scale caused by different surface and backscatter conditions between the specific Tan-DEM-X acquisitions (Fig. S2b). However, due to the limited extent of overlapping glacier areas (Fig. S2a), it is not possible to derive a date-specific intensity correction for each DEM strip. Thus, the applied linear model does rather represent an average difference in surface penetration depth between autumn and winter SAR data."

Accepted.

\*The intensity images are created using the Gamma remote sensing software environment (Werner et al., 2000). The radiometric calibration of the amplitude to  $\sigma$ 0 values is automatically performed by the conversion algorithm from the CoSSC to the Gamma data format (using the metadata of the CoSSC data product). The respective algorithms are part of the interferometry (ISP) module and described in the Interferometric SAR Processor – ISP user's guide (GAMMA Interferometric SAR Processor (ISP), 2021) in section 2.2.7 (TerraSAR-X & TanDEM-X data read algorithms) and 2.4.5 (radiometric calibration procedure). A link to this user guide is provided in the reference list.

Still the backscattering values used for the signal penetration (Fig 1c and Fig 1d) are going down to -27 dB. The noise level (NESZ) for the beams around 37-41 deg incidence is around -24 dB. It is annotated in each CoSSC product. This questions additionally EQ2.

 There are some studies which showed similar radar measurements below the noise level, e.g. (Meng et al., 2017) showed values of -30 db and less for low-scattering (ocean) areas. We assume that the fraction of September acquisitions with relatively low backscatter values are also such low-scattering areas and not caused by an error in the DEM creation. Nevertheless, the linear regression would be almost identical if we would include some of the very low backscatter values (see Fig. 1d).

\*Concerning the incidence angle, we revised Table S2 following the suggestions (see respective comment below) and extended the methods section (see second comment).

Accepted.

\*The hypsometric bars shown in Fig. 2 refer to the normalized median absolute deviation of  $\Delta h/\Delta t$ measurements on glacier areas within each elevation bin. Therefore, the bars are largest a low elevations because the spread of measured  $\Delta h/\Delta t$  values is large due to the presence of strong thinning glacier termini. At high altitudes, the range of measured  $\Delta h/\Delta t$  values is in general much smaller (see also  $\Delta h/\Delta t$  maps of Fig.2) and thereby also the bar. We extended the caption of Fig. 2 because the description of the shown bars was missing. Regarding the geodetic error, we did not calculate a mass change error for each elevation bin but for the entire region (based on the mean regional elevation change and respective uncertainty). The errors in dh/dt [m/yr] related to vertical co-registration do not depend on the magnitude of retrieved dh/dt, but on the uncertainty in co-registration, independent of the magnitude. Even zero dh/dt has the same error in respect to vertical co-registration. For the higher elevation zones (firn areas) where penetration-related errors are added, the error in the elevation change rate dh/dt [m/yr] should by higher in the firn areas than in ice areas of glaciers. Fig. 2 d to f: Please explain the term "normalized median absolute deviation of elevation change measurements of each elevation bin". An equation would be helpful.

- Agree, the magnitude of the specific dh/dt error of an elevation bin is related to the respective accuracy of the co-registration and DEM data. In most cases, this error increases at high altitudes with steep slopes and a rugged topography. However, the "error bars" shown in Fig. 2 do not refer to the actual elevation change error but indicate the range of different glacier elevation change values within each elevation bin.
- The size of the bars in Fig. 2d-f was simply derived by calculating the normalized median absolute deviation (NMAD) of glacier dh/dt values of each elevation bin. Thereafter the NMAD was subtracted/added to the mean glacier dh/dt value of the respective elevation bin. We selected this statistical measure to indicate the spread of elevation change values of each elevation bin which contributed to the mean change value. We used the NMAD instead of showing the minimum and maximum elevation change value of each bin because otherwise the size of bars would be very large at some elevations which would decrease the visibility of the overall hypsometric distribution of the mean change values.

\*Regarding to Eq. 1 (Supplement), unfortunately we do not quite understand the question referring to the mass change and elevation change uncertainty: The first term of the sum is the ratio between the  $\Delta h/\Delta t$  uncertainty ( $\delta \Delta h/\Delta t$ ) and the mean (glacier)  $\Delta h/\Delta t$  estimate. While the  $\Delta h/\Delta t$  estimate is derived on glacierized areas, the  $\Delta h/\Delta t$  uncertainty ( $\delta \Delta h/\Delta t$ ) mainly indicates the potentially remaining offsets on non- glacier areas after the co-registration (and also other sources of uncertainty, Supplement Eq.2). If  $\Delta h/\Delta t$  would be very small (and thereby also  $\Delta M/\Delta t$ ), the uncertainty of  $\Delta M/\Delta t$  could still be relatively high if  $\delta \Delta h/\Delta t$  (off-ice) is high compared to  $\Delta h/\Delta t$  (on-ice). For example, the measured elevation change rate is rather small but there are a lot of artificial elevation offsets remaining after the co-registration. In this case  $\delta \Delta M/\Delta t$  would be high compared to  $\Delta M/\Delta t$ .

Accepted. For Eq. 1 (Supplement) there was some misunderstanding because of the same symbol (delta) used for relative error (right hand side) and absolute error (left hand side). The error estimate refers to the total change of mass over an extended area (though not defined; should be explained). There is one (rather unlikely case), when the equation is wrong: if the retrieved 2h/2t is exactly zero. According to Eq. 1 this yields zero error for the mass change.

\*Fig. 1a shows a subset of backscatter values (5000 random samples) because otherwise the figure would be too busy. The acquisitions of December 2016 cover only a very small fraction of the Novaya Zemlya ice cap (~10 km2). For this reason, there are only very few December datapoints visible and the mean backscatter was only calculated and plotted for the lowest elevation bin. The mean value is almost the same as for October 2016 (triangle) and therefore difficult to identify in the figure.

Now I can see it. Thanks for the explanation!

\*We changed Table S2 and included only one row per TanDEM-X acquisition instead of each CoSSC frame. The terminology was adjusted and the new columns include now, as suggested, acquisition date, acquisition start time, active satellite, orbit direction, relative orbit, strip length (number of CoSSC frames), effective baseline, height of ambiguity and incidence angle.

Accepted.

**References:**

Dall, J.: InSAR elevation bias caused by penetration into uniform volumes, IEEE Trans. Geosci. Remote Sens., 45, 2319–2324, 2007

Ulaby, Fawwaz & Long, David & Blackwell, William & Elachi, Charles & Fung, Adrian & Ruf, Christopher & Sarabandi, K. & Zyl, Jakob & Zebker, Howard. (2014). Microwave Radar and Radiometric Remote Sensing.

MARCHENKO, S., POHJOLA, V., PETTERSSON, R., VAN PELT, W., VEGA, C., MACHGUTH, H., . . . ISAKS-SON, E. (2017). A plot-scale study of firn stratigraphy at Lomonosovfonna, Svalbard, using ice cores, borehole video and GPR surveys in 2012–14. Journal of Glaciology, 63(237), 67-78. doi:10.1017/jog.2016.118

**References new:**

Dall, J.: InSAR Elevation Bias Caused by Penetration Into Uniform Volumes, IEEE Trans. Geosci. Remote Sens., 45, 2319–2324, https://doi.org/10.1109/TGRS.2007.896613, 2007.

Kravchenko, I., Besson, D., and Meyers, J.: In situ index-of-refraction measurements of the South Polar firn with the RICE detector, J. Glaciol., 50, 522–532, https://doi.org/10.3189/172756504781829800, 2004.

Marchenko, S., Pohjola, V. A., Pettersson, R., Van Pelt, W. J. J., Vega, C. P., Machguth, H., Bøggild, C. E., and Isaksson, E.: A plot-scale study of firn stratigraphy at Lomonosovfonna, Svalbard, using ice cores, borehole video and GPR surveys in 2012–14, J. Glaciol., 63, 67–78, https://doi.org/10.1017/jog.2016.118, 2017.

Meng, H., Wang, X., Chong, J., Wei, X., and Kong, W.: Doppler Spectrum-Based NRCS Estimation Method for Low-Scattering Areas in Ocean SAR Images, Remote Sens., 9, 219, https://doi.org/10.3390/rs9030219, 2017.

---

## Author Response (AR4)

**Comments to the author:**

Dear Authors,

You considered carefully the last comments of the referee and I am happy to accept your manuscript for publication.

However, before final publication and copy-editing, I ask you to take into account carefully my comments below. There are still a few typos in the text and some statements need further clarifications. In particular, a non specialist like me has a bit of a hard time to make sense of the different values quoted for the penetration depth of the radar signal.

To facilitate and speed-up my assessment (I do not want to delay further publication), I to ask you to upload a point-by-point response, the revised manuscript both cleaned and track-changed.

Best regards,

Etienne Berthier

**Thank you very much, we made some corrections according to the suggestions, did some proofreading and extended the description of the different signal penetration values in the supplement. Please find our point-by-point responses below.**

**Best regards,**

**Christian Sommer**

L42: "over most of our study area" sounds better to me (the study area has already been defined just above)

**\*Agree, changed to "over most of our study area"**

L47 "respective" not needed

**\*Ok, deleted**

L66. in "beta_1", 1 should be a subscript (like done for 0)

**\*Ok, corrected subscript**

L67. Does "autumn" corresponds to September? Not necessarily obvious (officially "autumn" starts 21 September). Maybe stick to September (everywhere in the text), or define "autumn".

**\*Agree, the phrase "autumn" corresponded to September (in most cases) but it is probably less confusing to use "September" everywhere. We replaced "autumn" with "September" whenever a sentence specifically referred to September data.**

L108. I find "measured" ambiguous. I think "uncorrected" (as you defined it above) would be best.

**\*Agree, it should be "uncorrected" and not "measured"**

L134. I think I would rather write "the occurrence of melt or presence of fresh snow" (presence of melt does not sound right or rather "presence of meltwater")

**\*Ok, changed accordingly**

L150. See my comment above about "autumn", a bit ambiguous. Check everywhere.

**\*Agree, replaced "autumn" in most cases**

L169. "smaller" is not clear. Do you want to say "less negative"? And then "less negative" than what? To be improved. Maybe quote the value of the mass change rate you want to compare to (the sum for the 4 glaciers you listed above, maybe?).

**\*This sentence should compare the regional mass change rate of entire Severnaya Zemlya and Severnaya Zemlya without the (4) most changing glaciers. Our total mass change result for Severnaya Zemlya is more negative compared to other studies but most of the mass loss is confined to those 4 glaciers, e.g. the Vavilov ice cap had a large surge event within the TanDEM-X observation period. Therefore, we also wanted to provide a number for the remaining majority of glacier areas which show much smaller mass changes. Rephrased sentence to*: "For the remaining glacierized areas of the Severnaya Zemlya Archipelago, the mass change rate is much smaller (-2.39 Gt $a^{-1}$, 850 kg $m^{-3}$) than for the entire region (-4.70 Gt $a^{-1}$, 850 kg $m^{-3}$)."**

L180. A thought (not necessarily leading to changes in the MS). Can you exclude a melt/refreezing event between your two set of measurements (for example in summer 2015 or 2016) that would create an efficient scattering surface not so deep in the firn and would induce this apparent uplift ?

**\*No, we cannot entirely exclude this possibility but the radar metadata does not show clear indications for signal penetration at these areas. Also, the locations of glacier areas which show those small gains are very similar to the elevation change maps shown by the referenced altimetry studies.**

Figure 1a : I could not find the curve for the 2016-12 backscatter. Were these data included? Maybe I missed them, sorry of this is the case.

**\*The 2016-12 data is in fact difficult to find in Fig. 1a because in December 2016 only a very small area of Novaya Zemlya was covered. This is due to the irregular acquisition plan of the TanDEM-X satellites. In 2016-12 only one DEM strip was acquired which covered some parts of the northern coastline but not the interior of the Novaya Zemlya ice cap. Also, we could only show a subset of sample points in the figure. Therefore, very few datapoints of December are visible at low elevations (<100 m a.s.l.). Nevertheless, I added "In December 2016 (blue crosses), only a small glacier area at the Northeastern coast was acquired." to the caption because readers might also have difficulties to find those datapoints.**

Fig 1d: maybe I missed something but the mean difference in signal penetration in the cruve appears much larger than the ~2 m values quoted in the text (rather around 3 m here on average). See also comments below on the values in the supplement. In the end a reader would have a hard time quoting a mean penetration depth from your study.

**\*The different measures of signal penetration in literature can be sometimes confusing because in many cases the penetration depth is defined as the vertical bias between two surfaces. This is often the case when comparing two different elevation datasets. However, the actual length of penetration into a (glacier) volume can be obviously much higher due to the side looking geometry of the radar antennas and the surface topography (e.g. penetration into a flat surface versus a steep slope). The approx. 2 m stated in the text are the vertical difference between the September und winter surface. In detail, the 2.13 m are the vertical difference on the overlapping glacier areas, which we can measure by differencing the respective DEMs, while the ~2.3 m are the modelled vertical difference for all September DEMs (estimated by the linear regression).**

**The values shown in Figure 1d are higher because those are the estimated penetration lengths and not the vertical offsets (of overlapping glacier areas). We extended the description of the different vertical and penetration length values in the supplement (see last comment).**

Supplement

L132. Add space between 150 and "m"

**\*Ok**

L136. Can you help the non SAR specialist reader and compare these numbers with the values of the main text which are 2.13 m (L100) and 2.3 m (L108). In the end it is a bit confusing.

**\*Agree, we rephrased this paragraph and extended the description of vertical elevation offsets and different surface penetration estimates (see comment below).**

L140. Penetration of 5.4 m or 3.1 m (or 2.13 m in the text) seems to translate in a larger difference over a total of 6 years. Maybe clarify why the corrections are so similar.

*Both methods (supplement sections 3.1. & 3.2.) use the vertical offsets (Winter – September elevation) and topography (local incidence angle) as input variables. (For the two-way power penetration (3.2.), the local incidence angle has to be converted additionally to the refraction angle but this introduces only very small changes.) The spatial distributions of the estimated penetration lengths of both methods are rather similar but the magnitude is different (3.1 m versus 5.4 m). However, these differences in magnitude do not have much impact on the derived linear regressions as the general correlation between backscatter and surface penetration difference is the same.

Eventually, both equations are rearranged to convert the modelled surface penetration lengths of all September areas back into vertical offsets. This is necessary to correct the elevation change rate. In the end, the actual estimated vertical offset is very similar because both linear regressions are based on almost the same input data.

The respective paragraph in the supplement was extended and summarizes now all values (L.136-146):

*"Eventually, the mean signal penetration length $l_p$ of the two-way power penetration conversion is 5.4 m while the trigonometric estimate is 3.1 m. In general, both estimates are higher than the measured vertical elevation difference of overlapping glacier areas $\Delta h_{W-A}$ (2.13 m) due to the side-looking geometry of the TanDEM-X SAR sensor. However, the spatial distribution of penetration $l_p$ and the derived linear regressions (see 2.1) are similar because both conversions are based on the measured vertical offset and local incidence angle. Therefore, when rearranging Eq. 1 and Eq. 2, the influence on the actual spatial vertical correction of glacier areas which were acquired during September 2016 is very small. The average vertical correction values for all September 2016 glacier areas are 2.29 m and 2.30 m, respectively. Eventually, the corrected elevation change rate of Novaya Zemlya ($\Delta h/\Delta t_{corr.}$) is less than 0.01 m $a^{-1}$ more negative when using the two-way power penetration estimate and the geodetic mass change results calculated with both values are almost identical."*